# Pretraining a Shared Q-Network for Data-Efficient Offline Reinforcement Learning

**Jongchan Park**[*]
Hyundai Motor Company
jcpark11@hyundai.com

**Mingyu Park**[*]
KAIST
m1n9yu@kaist.ac.kr

**Donghwan Lee**
KAIST
donghwan@kaist.ac.kr

## Abstract

Offline reinforcement learning (RL) aims to learn a policy from a fixed dataset without additional environment interaction. However, effective offline policy learning often requires a large and diverse dataset to mitigate epistemic uncertainty. Collecting such data demands substantial online interactions, which are costly or infeasible in many real-world domains. Therefore, improving policy learning from limited offline data—achieving high data efficiency—is critical for practical offline RL. In this paper, we propose a simple yet effective plug-and-play pretraining framework that initializes the feature representation of a $Q$-network to enhance data efficiency in offline RL. Our approach employs a shared $Q$-network architecture trained in two stages: pretraining a backbone feature extractor with a transition prediction head; training a $Q$-network—combining the backbone feature extractor and a $Q$-value head—with *any* offline RL objective. Extensive experiments on the D4RL, Robomimic, V-D4RL, and ExoRL benchmarks show that our method substantially improves both performance and data efficiency across diverse datasets and domains. Remarkably, with only **10%** of the dataset, our approach outperforms standard offline RL baselines trained on the full data.

## 1 Introduction

Sample efficiency is a long-standing challenge in reinforcement learning (RL). Typical RL algorithms rely on an online learning process that alternates between collecting experiences through interactions with the environment and improving the policy [55]. However, acquiring a large number of online interactions is often impractical, since data collection can be costly and risky. Offline reinforcement learning (RL) has emerged as a promising alternative to address this issue by decoupling data collection from policy learning, enabling agents to learn solely from pre-collected datasets [37]. Learning an offline policy from a static dataset allows the agent to focus on how effectively it can extract optimal behaviors from a fixed data distribution—unlike online RL, which must continuously expand its experience data through active exploration. Nevertheless, learning an optimal policy from limited experience data remains a fundamental challenge in both online and offline RL.

However, prior offline RL approaches have largely focused on improving policy learning within a fixed dataset through policy constraints [18, 32], conservative regularization [33], and model-based uncertainty estimation [28] to mitigate distributional shift. Other studies, such as data-manipulation strategies [27, 67, 61] and offline-to-online frameworks [43, 59, 47, 3], provide alternative perspectives on improving policy performance with limited data sources. However, understanding how an offline agent learns an effective policy with minimal data usage offers a distinct perspective. To address this, we define *data efficiency* as the ability of an offline RL agent to learn an optimal policy from as little offline data as possible, which is distinct from sample efficiency in online RL. Furthermore, we consider truly *data-efficient* offline RL as learning an optimal policy across datasets

---

[*]Equal Contribution. Correspondence to donghwan@kaist.ac.kr.

39th Conference on Neural Information Processing Systems (NeurIPS 2025).

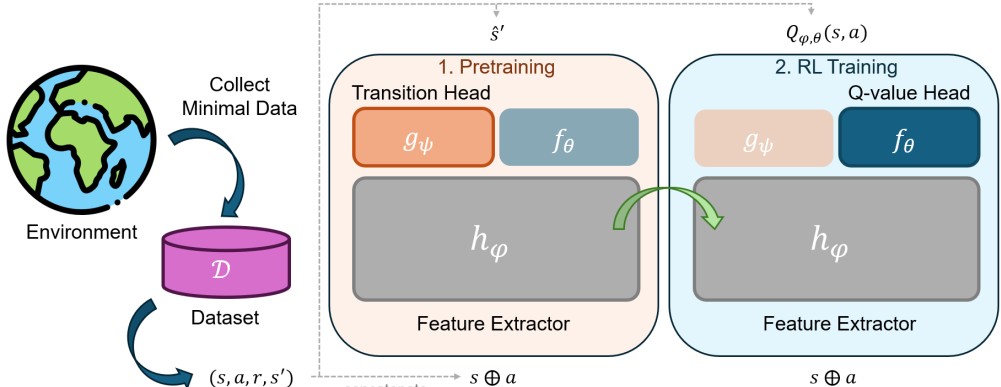

Figure 1: **Overview of the pretraining framework.** Our approach decomposes the original $Q$-network into two core architectures: a shared backbone network that extracts the representation $z$ from the concatenated state-action input $(s, a)$, and two shallow head networks for learning the transition model and estimating $Q$-values, respectively.

that vary in size, coverage, and behavioral optimality, rather than focusing on performance within a single fixed dataset.

In this work, we propose a simple yet effective plug-and-play framework that pretrains a shared $Q$-network via a two-stage learning strategy toward data-efficient offline RL. As illustrated in Figure 1, our shared $Q$-network consists of a backbone feature extractor $h_\varphi$ and two shallow head networks: $g_\psi$ for next-state prediction and $f_\theta$ for $Q$-value estimation. In the pretraining stage, we train $h_\varphi$ jointly with $g_\psi$ through a transition prediction task, encouraging the backbone to encode dynamics-relevant representations. In the subsequent RL training stage, we fine-tune $h_\varphi$ alongside $f_\theta$ using *any* standard offline RL objective. This modular design enables seamless integration with existing offline RL algorithms while improving representation quality for value estimation.

We theoretically analyze the effect of such pretraining using the projected Bellman equation under linear function approximation. Our analysis reveals that initializing the feature matrix (backbone feature extractor) with pretrained representations increases its rank, which tightens the upper bound on the optimal $Q$-value estimation error. Consequently, the pretrained $Q$-network achieves faster and more accurate convergence than conventional methods. Empirically, this structural property is validated by observing higher feature matrix ranks and lower $Q$-value proxy errors, as shown in Figure 3 and Table 1.

Finally, extensive experiments demonstrate that our approach substantially enhances both the performance and data efficiency of existing offline RL algorithms across diverse benchmarks, including D4RL [15], Robomimic [40], V-D4RL [38], and ExoRL [62]. Our method maintains strong performance even with limited data subsets and under varying data qualities and collection strategies. Notably, with only **10%** of the dataset, our pretrained $Q$-network outperforms standard baselines trained on the full dataset. Moreover, our method surpasses both offline model-based and representation learning approaches on reduced datasets, confirming its general effectiveness in data-efficient offline policy learning.

Further discussions on how data efficiency in offline RL differs from sample efficiency in online RL are in Appendix A. Additionally, the source code is available in our GitHub repository[2]. With only a few additional lines on top of the original TD3+BC[3] implementation, we demonstrate that our method is simple to implement and easily adaptable to other offline RL algorithms.

## 2  Related Works

**Offline RL.** Offline RL aims to learn policies solely from a fixed dataset without further interaction with the environment. A major challenge in this setting is distribution shift, where queries to

---

[2]`https://github.com/daisophila/PSQN.git`
[3]`https://github.com/sfujim/TD3_BC`

the $Q$-function on out-of-distribution actions can lead to overly optimistic value estimates during training [18, 32, 37, 33, 16, 29]. Recent studies have explored scaling offline RL algorithms to larger datasets and model capacities [9, 44, 56], as well as offline-to-online RL paradigms that pretrain agents offline before fine-tuning them online to enhance sample efficiency [43, 59, 47, 3].

Beyond these standard formulations, other works have investigated diverse data conditions, such as imbalanced or corrupted datasets and the use of unlabeled data within the offline RL framework [27, 67, 61]. Although several studies [1, 30, 33] have evaluated performance on data subsets, few have explicitly addressed *data efficiency*—that is, how well an offline RL agent can learn with minimal data. In contrast, our work directly targets this problem. We propose a simple yet effective plug-and-play pretraining method that enhances data efficiency by initializing a shared $Q$-network for improved policy learning from limited static datasets.

**Sample-Efficient RL.** A persistent challenge in most RL algorithms is sample inefficiency, as learning optimal policies typically requires extensive online interactions. Addressing this limitation has been a long-standing focus of RL research [65, 66, 12]. One prominent approach is model-based RL, which improves efficiency by learning a (possibly latent) dynamics model to generate additional synthetic transitions [54, 11, 22, 21, 25]. Alternatively, techniques such as representation pretraining [50, 51, 66] and data augmentation [34, 65] have shown strong empirical gains in sample efficiency by enhancing feature reuse and robustness.

More recently, offline-to-online RL [36, 3, 47, 14, 43] and foundation model approaches [2, 52, 7, 6, 4] have emerged to mitigate the poor sample efficiency of online RL. These methods leverage large-scale offline data or pretrained models to accelerate subsequent online adaptations, underscoring the growing convergence between sample-efficient and data-driven RL paradigms.

**Data-Efficient Offline RL.** In this work, we define *data efficiency* in offline RL as the ability of an algorithm to learn an optimal policy from a minimal set of pre-collected samples. This differs from sample efficiency in online RL, which concerns minimizing environment interactions. While prior works [50, 51] have discussed "data-efficient" RL, their primary focus was on online settings—thus addressing sample efficiency rather than true offline data efficiency. These methods typically rely on self-predictive representation learning in latent spaces, often combined with techniques like data augmentation [65] or momentum target encoders [26].

By contrast, our method employs self-supervised pretraining within a shared network architecture to improve representation quality, without requiring auxiliary techniques or additional data transformations. Through extensive experiments under various dataset qualities and distributions, we demonstrate that our approach consistently improves performance in offline RL, effectively addressing the data efficiency problem as defined in this work.

In Appendix B, we provide additional discussions on related approaches and broader connections to representation learning and model-based methods in RL.

## 3 Pretraining Q-network with Transition Prediction Improves Data Efficiency

In this paper, we propose a simple yet effective pretraining framework that transfers learned transition features into the initialization of $Q$-network to improve data efficiency in offline RL. To this end, we design a shared $Q$-network architecture combining a backbone feature extractor $h_\varphi$ and two shallow head networks: a transition head $g_\psi$ for next-state prediction and a $Q$-value head $f_\theta$ for estimating $Q$-value. We further introduce a **two-stage learning strategy**—a pretraining and an RL training—built upon the shared $Q$ network for data-efficient offline RL. During the pretraining stage, the transition model $g_\psi \circ h_\varphi$ predicts the next state given state-action pairs $(s, a)$:

$$\hat{s}' = (g_\psi \circ h_\varphi)(s, a), \quad (s, a) \in \mathcal{S} \times \mathcal{A}, \tag{1}$$

where $\hat{s}'$ denotes the predicted next state, and $g_\psi$ is a parameterized linear function.

In the subsequent RL training stage, the same backbone network $h_\varphi$ is shared with the $Q$-value head $f_\theta$, forming the $Q$-network:

$$Q_{\varphi,\theta}(s, a) = (f_\theta \circ h_\varphi)(s, a), \quad (s, a) \in \mathcal{S} \times \mathcal{A}, \tag{2}$$

where $f_\theta$ represents a linear output layer, and $h_\varphi$ corresponds to the fully connected layers shared with the transition model in (1). The overall shared architecture is illustrated in Figure 1.

---
**Algorithm 1** Pretraining a shared Q-network scheme for offline RL
---
1: **Input**: Dataset $\mathcal{D}$ of transition $(s, a, s')$, learning rate $\alpha$, initialized parameters $\varphi, \psi$
2: **for** each gradient step **do**
3:     Sample a mini-batch $\mathcal{B} \sim \mathcal{D}$
4:     Compute the next-state prediction error

$$\mathcal{L}_{pre}(\varphi, \psi) = \sum_{(s,a,s') \in \mathcal{B}} (s' - (g_\psi \circ h_\varphi)(s, a))^2$$

5:     Update the weights of the backbone feature extractor and the transition prediction head of the shared network

$$\varphi \leftarrow \varphi - \alpha \nabla_\varphi \mathcal{L}_{pre}(\varphi, \psi), \quad \psi \leftarrow \psi - \alpha \nabla_\psi \mathcal{L}_{pre}(\varphi, \psi)$$

6: **end for**
7: **Output**: Pretrained weights $\varphi$ of the backbone feature extractor of the shared network
---

We pretrain the transition model $g_\psi \circ h_\varphi$ via self-supervised regression by minimizing the mean-squared prediction error:

$$\mathcal{L}_{pre}(\varphi, \psi) = \sum_{(s,a,s') \in \mathcal{D}} \|s' - (g_\psi \circ h_\varphi)(s, a)\|_2^2 \tag{3}$$

where $\mathcal{D}$ denotes the static dataset of transition tuples $(s, a, s')$.

After pretraining, the parameters $\varphi$ can be fine-tuned or frozen when training standard RL algorithms using the $Q$-network structure above, without requiring any architectural modification. By default, we fine-tune the backbone feature extractor during the RL training stage, and report results with a frozen backbone in Appendix F. The complete pretraining procedure is summarized in Algorithm 1.

### 3.1 Analysis Based on the Projected Bellman Equation and a Proxy Q Error

In this section, we analyze how our method improves *data efficiency* through the lens of the projected Bellman equation. For clarity, we assume discrete and finite state–action spaces with deterministic transitions. However, the core principles naturally extend to continuous domains.

Our analysis begins by noting that the $Q$-function parameterized by neural networks can be expressed as in (2). We decompose the network into two parts: a feature extractor $h_\varphi$ and a linear function approximator $\theta$. Letting $z = h_\varphi(s, a) \in \mathbb{R}^m$, the $Q$-function can be rewritten as $Q_{\varphi,\theta}(s, a) = \sum_{i=1}^{m} \theta_i h_{\varphi,i}(s, a) = \langle \theta, h_\varphi(s, a) \rangle$ where $(s, a) \in \mathcal{S} \times \mathcal{A}$.

When $\varphi$ is fixed, then the above structure can be viewed as a linear function approximation with the feature function $h_{\varphi,i}$. Our method pretrains $h_{\varphi,i}$ by minimizing the prediction loss in (3). Here, the latent feature $z$ corresponds to the MLP output before the final layer, while $\theta$ parameterizes the linear output layer; i.e., $Q_{\theta,\varphi}(s, a) = h_\varphi(s, a)^T \theta$. Thus, interpreting our network under the linear approximation framework provides a useful model to explain its improved data efficiency.

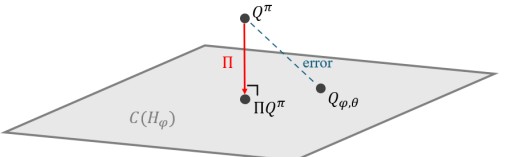

Figure 2: **Reduced approximation error through the expanded column space of $H_\varphi$.** In linear approximation, the true value function $Q^\pi$ may lie outside the column space of $H_\varphi$. The projected Bellman equation addresses this by projecting $Q^\pi$ onto its closest representation $\Pi Q^\pi$ within the column space of $H_\varphi$.

It is well known that under linear function approximation, the standard Bellman equation

$$Q_{\varphi,\theta}(s, a) = R(s, a) + \gamma \sum_{s' \in S} P^\pi(s'|s, a) \sum_{a' \in A} Q_{\varphi,\theta}(s', a')$$

may not admit a solution in general. However, typical TD-learning algorithms are known to converge to the unique fixed point of the projected Bellman equation [41]. In particular, considering the vector

form of the Bellman equation, $Q_{\varphi,\theta} = R + \gamma P^\pi Q_{\varphi,\theta}$, the projected Bellman equation is known to admit a solution

$$Q_{\varphi,\theta} = \Pi(R + \gamma P^\pi Q_{\varphi,\theta})$$

where $\Pi$ denotes the projection operator onto the column space $C(H_\varphi)$ of the feature matrix $H_\varphi$ defined as

$$H_\varphi := \begin{bmatrix} \vdots \\ h_\varphi(s,a)^T \\ \vdots \end{bmatrix}.$$

The corresponding approximation error is bounded by

$$||Q_{\varphi,\theta} - Q^\pi||_\infty \leq \frac{1}{1-\gamma}||\Pi Q^\pi - Q^\pi||_\infty, \tag{4}$$

where $Q^\pi$ is the true $Q$-function corresponding to the target policy $\pi$. This bound highlights that the estimation error depends on the expressiveness of $H_\varphi$. A richer feature representation—i.e., a column space $C(H_\varphi)$ that better spans $Q^\pi$—leads to a smaller Bellman error (Figure 2).

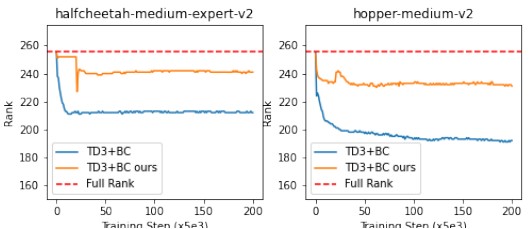

Figure 3: **The Rank of the latent space in the $Q$-network during training.** We compare the rank of the latent representations between vanilla TD3+BC and TD3+BC+Ours using 512 samples. Our method consistently maintains a higher latent-space rank, indicating a richer feature representation and reduced approximation error.

To empirically validate this interpretation, we compare the rank of the latent feature space between the vanilla TD3+BC and our pretrained TD3+BC models using 512 samples. Following [31], we measure the hard rank (instead of soft rank used in [31]). As shown in Figure 3, our method yields significantly higher feature-space rank, indicating that it expands $C(H_\varphi)$ and thus captures a larger subspace of $\mathbb{R}^{|\mathcal{S}\times\mathcal{A}|}$. This expansion enables more precise $Q$-function estimation with the same number of samples—equivalently, more accurate learning with less data.

To further verify that our method reduces Bellman error under limited data, we define a proxy $Q$-error based on the left-hand side of (4). We estimate $Q^\pi$ using the critic of TD3+BC trained with the full dataset, and estimate $Q_{\varphi,\theta}$ from TD3+BC and TD3+BC+Ours trained with only 10% of the data. Across D4RL benchmarks, Table 1 shows that our method consistently yields lower proxy $Q$-error than the baseline. These results demonstrate that our pretrained representation facilitates more accurate $Q$-function estimation, leading to superior data efficiency in offline RL.

Table 1: **Comparison of proxy $Q$-error between TD3+BC and TD3+BC+Ours.** We compare the proxy $Q$-error of vanilla TD3+BC and TD3+BC+Ours, both trained on 10% of the D4RL datasets. The reference optimal $Q$-value is estimated using the critic of TD3+BC trained on the full datasets. Our method consistently achieves lower proxy $Q$-error, demonstrating more accurate $Q$-function estimation with substantially less data—highlighting its superior data efficiency.

|  |  | TD3+BC | TD3+BC + Ours |
|---|---|---|---|
| Medium | HalfCheetah | 503.6851 | **287.7740** |
|  | Hopper | 536.2078 | **472.9333** |
|  | Walker2d | 299.1773 | **138.0320** |
| Medium Replay | HalfCheetah | 1032.6671 | **119.9188** |
|  | Hopper | 320.1790 | **319.9531** |
|  | Walker2d | 379.4456 | **49.9275** |
| Medium Expert | HalfCheetah | 522.4211 | **112.2337** |
|  | Hopper | 437.8861 | **254.9155** |
|  | Walker2d | 392.7360 | **148.2040** |

# 4 Experiments

Table 2: **Average normalized scores on the D4RL benchmark.** Each column corresponds to a different RL baseline. The values on the left represent the baseline scores reported in the original literature, while the values on the right show the results of our method combined with each baseline. Performance improvements over the original baselines are highlighted in blue. All results are reported with the mean and standard deviation scores over five random seeds.

| | | AWAC | CQL | IQL | TD3+BC |
|---|---|---|---|---|---|
| Random | HalfCheetah | 2.6±0.4→51.1±0.9 | 21.7±0.9→31.9±2.6 | 10.3±0.9 → 18.3±1.0 | 2.3±0.0→14.8±0.5 |
| | Hopper | 28.6±8.9→59.5±33.8 | 10.7±0.1→30.2±2.7 | 9.4±0.4 → 10.7±0.4 | 10.7±3.2→31.6±0.2 |
| | Walker2d | 7.8±0.2→13.1±3.9 | 2.7±1.2→19.6±4.5 | 7.9±0.5 → 8.9±0.7 | 5.9±1.0→11.2±5.1 |
| Medium | HalfCheetah | 48.4±0.1→54.6±1.5 | 37.2±0.3→39.9±18.8 | 46.6±0.2→48.9±0.2 | 43.5±0.1→49.2±0.3 |
| | Hopper | 88.4±8.8→101.7±0.2 | 44.2±10.8→90.6±2.2 | 76.9±5.8→78.6±2.2 | 67.0±1.9→71.5±2.2 |
| | Walker2d | 53.0±33.2→89.5±0.9 | 57.5±8.3→84.7±0.7 | 83.8±1.5→83.6±1.1 | 82.1±1.0→87.1±0.6 |
| Medium Replay | HalfCheetah | 46.1±0.3→55.8±1.3 | 41.9±1.1→47.6±0.4 | 43.4±0.3→45.5±0.2 | 40.0±0.5→45.8±0.3 |
| | Hopper | 101.3±0.6→106.7±0.6 | 28.6±0.9→98.6±2.1 | 96.2±1.9→99.4±1.7 | 73.9±7.3→100.2±1.6 |
| | Walker2d | 88.1±0.6→100.3±2.1 | 15.8±2.6→87.7±1.3 | 77.9±2.1→88.0±1.7 | 58.0±3.6→92.0±1.6 |
| Medium Expert | HalfCheetah | 76.4±2.8→90.1±1.9 | 27.1±3.9→82.8±6.5 | 94.8±0.2→95.3±0.1 | 76.8±2.8→96.9±0.9 |
| | Hopper | 113.0±0.7→113.2±0.2 | 111.4±1.2→111.1±0.8 | 101.8±7.5→105.8±11.3 | 102.2±9.6→113.0±0.2 |
| | Walker2d | 103.3±15.3→111.9±0.3 | 68.1±13.1→91.6±42.5 | 111.6±0.7→112.1±0.9 | 109.5±0.2→111.6±0.4 |
| Expert | HalfCheetah | 94.4±0.8→93.5±0.1 | 82.4±7.4→97.1±1.0 | 96.4±0.2 → 97.4±0.1 | 94.0±0.2→98.9±0.6 |
| | Hopper | 112.8±0.4→112.9±0.1 | 111.2±2.1→112.1±0.4 | 113.1±0.6 → 113.3±0.5 | 113.0±0.1→113.4±0.3 |
| | Walker2d | 110.4±0.0→111.2±0.4 | 103.8±7.6→110.6±0.3 | 110.7±0.3 → 112.8±1.1 | 109.9±0.3→111.0±0.2 |

In this section, we evaluate the effectiveness of our method across a range of offline RL benchmarks, including standard D4RL, the more complex Robomimic domain, and the image-based V-D4RL environment. We further examine data efficiency by evaluating performance on partial subsets of D4RL and ExoRL datasets. We begin by describing the experimental setup and the baselines used for each experiment. The experimental evaluation is structured as follows: first, we compare performance improvements on standard offline RL benchmarks; second, we analyze data efficiency across varying dataset qualities; and finally, we investigate performance across different dataset distributions.

**Experimental setup and Baselines.** We consider heterogeneous tasks and diverse datasets to ensure comprehensive evaluation. For locomotion tasks, we evaluate our method on the D4RL benchmark [15] with three agents (*HalfCheetah, Hopper, Walker2d*) and five dataset types (*random, medium-replay, medium, medium-expert, expert*). Our method is applied to popular offline RL algorithms—AWAC [42], CQL [33], IQL [29], and TD3+BC [16]—and we compare normalized scores between the baseline and the baseline augmented with our pretraining framework.

For tabletop manipulation tasks, we use the Robomimic benchmark [40] with *Lift* and *Can* tasks and mixed-quality machine-generated (MG) datasets. We compare the success rates of IQL, TD3+BC, IRIS [39], and BCQ [18] with and without our method.

For high-dimensional vision-based tasks, we evaluate on *Cheetah Run* and *Walker Walk* in V-D4RL [38], building our method on top of DrQ+BC, which applies the same regularization of TD3+BC into DrQ-v2 [63].

For data-efficient offline RL, we investigate both the impact of dataset quality and dataset collection strategy. We evaluate reduced D4RL locomotion datasets on MOPO [68], MOBILE [53], and ACL [60], and reduced ExoRL datasets [62] (*walker walk: SMM [35], RND [8], ICM [46]; point mass maze: Proto [64], DIAYN [13]*) on TD3 [17] and CQL. Detailed experimental and implementation settings are provided in Appendix D and Appendix C.

## 4.1 Performance Improvement in Offline RL Benchmarks

To validate the effectiveness of our approach, we evaluate it on the D4RL and Robomimic benchmarks. Table 2 compares the normalized scores of baseline algorithms and their counterparts augmented with our method across various environments and datasets. When integrated with existing offline RL methods (*i.e.*, AWAC, CQL, IQL, and TD3+BC), our approach consistently improves performance in most settings. The blue-highlighted scores in Table 2 indicate gains over the corresponding baseline algorithms. Notably, AWAC shows an average performance improvement of $+140.37\%$ compared to its original version.

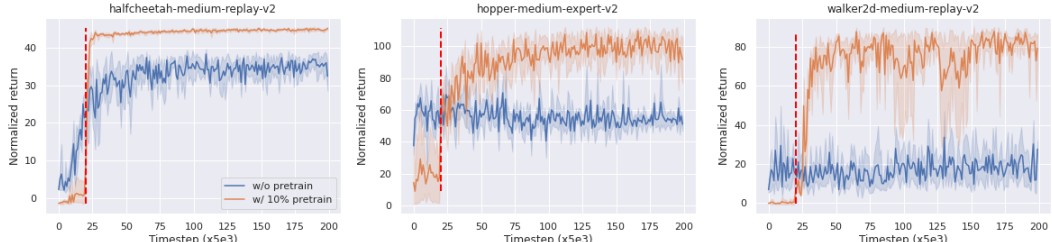

Figure 4: **Learning curves of TD3+BC.** We represent the normalized scores of the vanilla TD3+BC (blue) and TD3+BC (orange) with our pretraining method, respectively. The vertical red dashed lines indicate the transition between the pretraining and main training phases. After pretraining, TD3+BC with our method rapidly surpasses the vanilla baseline by a significant margin.

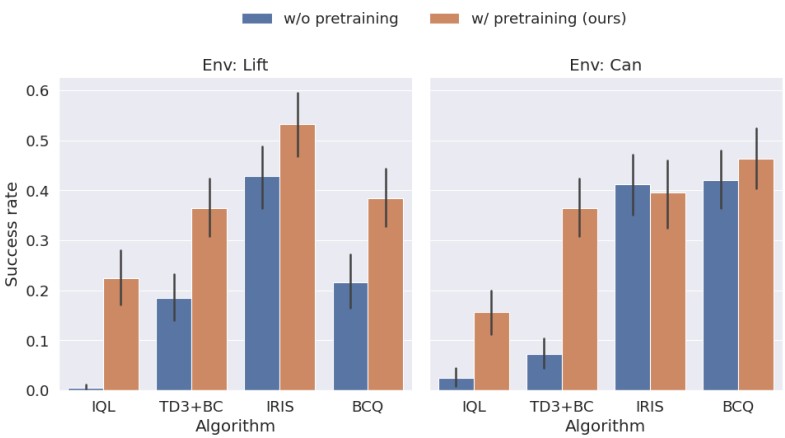

Figure 5: **Average success rates on the Robomimic benchmark.** We compare the baseline methods without pretraining (blue) against those augmented with our pretraining approach (orange) over three seeds. In seven out of eight tasks, our method yields a substantial improvement in success rate across both environments.

Figure 4 presents the learning curves of TD3+BC, demonstrating the clear benefit of our pretraining strategy. After the pretraining phase (denoted by the red vertical lines), the pretrained agent quickly outperforms the vanilla TD3+BC and achieves higher returns throughout training. These results indicate that our method accelerates convergence and enhances asymptotic performance with only minimal architectural or algorithmic modifications. Complete learning curves for TD3+BC are provided in Figure 12 of Appendix G.

We further evaluate our method on large-scale robotic manipulation tasks from the Robomimic benchmark to assess its effectiveness in complex, real-world scenarios. These tasks include suboptimal transitions, providing a challenging testbed beyond the D4RL benchmark. Figure 5 reports the averaged success rates of four offline RL baselines, both with and without our pretraining method. As shown, incorporating our approach consistently improves performance in seven out of eight cases, demonstrating its robustness in complex tasks.

Additional experiments on the Adroit benchmark (24-DOF control) are presented in Appendix E. We apply our method to AWAC, IQL, and TD3+BC across twelve settings (*i.e.,* four environments $\times$ three datasets) and evaluate each over five random seeds. In most cases, our method achieves clear performance gains, further confirming its effectiveness in high-dimensional control.

Finally, we examine the scalability of our approach to high-dimensional visual input using the V-D4RL benchmark [38]. Similar to other vision-based offline RL methods, our framework integrates seamlessly by replacing the state input with latent representations extracted from a visual encoder. As shown in Figure 3, our method consistently enhances the performance of DrQ+BC [63], validating its applicability to image-based environments.

Table 3: **Average episode returns on the V-D4RL benchmark.** We evaluate our approach in the image-based environment over three seeds. The results show that integrating our method consistently improves the performance of DrQ+BC.

|  |  | DrQ+BC | DrQ+BC + Ours |
|---|---|---|---|
| Medium | Walker walk | 306.93±28.21 | 338.77±29.55 |
|  | Cheetah Run | 340.33±7.55 | 379.80±45.83 |
|  | Humanoid Walk | 12.57±6.73 | 20.03±3.80 |
| Medium Replay | Walker walk | 30.09±0.75 | 28.68±2.29 |
|  | Cheetah Run | 21.15±2.04 | 25.13±2.04 |
|  | Humanoid Walk | 40.76±16.27 | 19.38±6.10 |
| Medium Expert | Walker walk | 352.46±37.15 | 369.66±20.86 |
|  | Cheetah Run | 251.52±34.37 | 258.76±50.33 |
|  | Humanoid Walk | 4.11±2.72 | 5.12±1.89 |

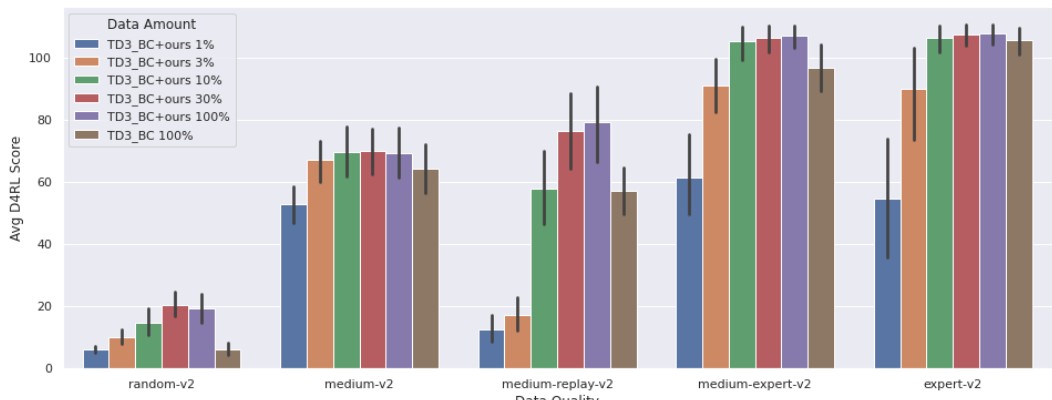

Figure 6: **Average normalized scores across varying dataset sizes and qualities.** We present the performance of our method on progressively reduced datasets *(1%, 3%, 10%, 30%, 100%)* across three D4RL environments: *HalfCheetah*, *Hopper*, and *Walker2d*. We demonstrate that our method remains highly data-efficient, achieving strong performance even with only 10% of the data— and as little as 1% for *random* datasets and 3% for *medium* datasets—regardless of data quality.

## 4.2 Data Efficiency across Data Qualities

To evaluate the data efficiency of our method across different dataset qualities, we tested it with TD3+BC on progressively reduced subsets of the D4RL datasets *(1%, 3%, 10%, 30%, and 100%)* spanning various data qualities *(random, medium, medium-replay, medium-expert, expert)*. Each reduced dataset was constructed by uniformly sampling transition segments $(s, a, r, s')$ from the full dataset, followed by both pretraining and RL training using these subsets.

As shown in Figure 14, our method demonstrates remarkable data efficiency. On the *random* datasets, training with only 1% of the data surpasses the performance of the vanilla TD3+BC trained on the full dataset for the *HalfCheetah* and *Walker2d* environments. Similarly, on the *medium* datasets, our method achieves comparable or higher performance with only 3% of the data. For higher-quality datasets (*medium-replay*, *medium-expert*, and *expert*), our method using merely 10% of the data consistently outperforms the vanilla TD3+BC trained on the entire dataset. Overall, as summarized in Figure 6, our method delivers robust performance with as little as 10% of the original dataset, confirming its strong data efficiency in offline RL.

We further compare our approach with representative offline model-based and representation-learning methods. Experiments are conducted on the *medium*, *medium-replay*, and *medium-expert* datasets of D4RL over three seeds. Figure 7 presents the aggregate results, while Figure 15 provides detailed comparisons. The results show that our method preserves high performance under reduced data conditions, unlike competing methods that incur additional training overhead (e.g., transition model

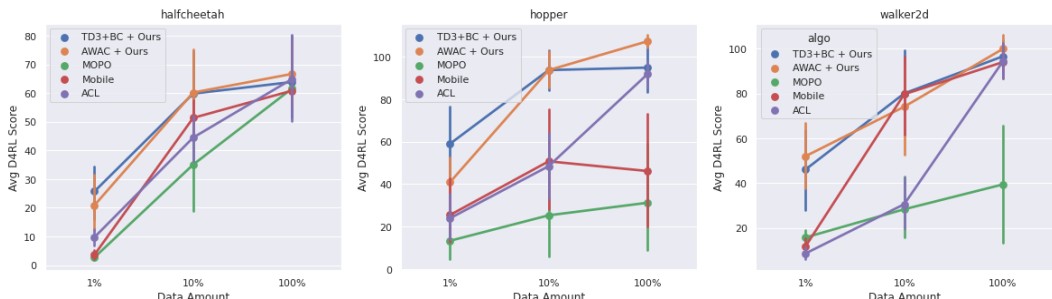

Figure 7: **Comparison with other approaches on the D4RL benchmark.** We compare our method against model-free offline RL baselines, model-based offline RLs (*MOPO*, *MOBILE*), and a representation learning approach (*ACL*) across three random seeds. The results are averaged over *medium*, *medium-replay*, and *medium-expert* datasets. Our method consistently maintains high performance under severe data reduction—particularly at **1%** of the dataset—where competing methods degrade significantly.

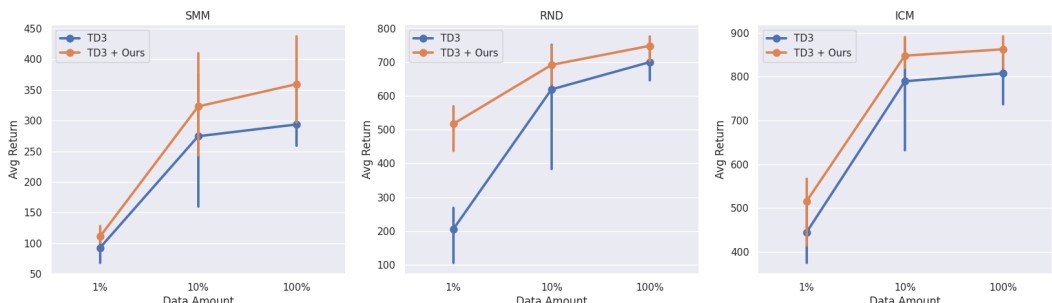

Figure 8: **Average episode returns in reduced datasets across data collection strategies.** We evaluate our method under different dataset collection strategies *(SMM, RND, ICM)*. Across all cases, TD3 combined with our method consistently outperforms vanilla TD3, even when trained with only 10% of the data—surpassing the performance of the full-data baseline. These results demonstrate that our method achieves strong data efficiency regardless of the underlying data distribution.

training and inference). Consequently, our method emerges as a more effective and computationally efficient choice for data-efficient offline RL.

## 4.3 Data Efficiency across Data Distributions

We hypothesize that smaller datasets induce distributional shifts compared to larger ones, as they often exhibit narrower coverage of the visited state space. To examine this effect, we evaluate our method across datasets generated by different collection strategies, each producing distinct data distributions. Using the ExoRL benchmark, we select TD3 as the baseline and consider datasets collected with SMM, RND, and ICM for the *walker walk* task. As reported in [62], ICM achieves the highest performance among the three, followed by RND and SMM. We compare vanilla TD3 and TD3 augmented with our method using reduced subsets of the datasets (*1%, 10%, 100%*) over three random seeds. Reduced datasets are constructed by taking the initial segments of the trajectories, and both pretraining and RL training are conducted on these subsets. As shown in Figure 8, our method consistently outperforms vanilla TD3 across all dataset types, even when using only 10% of the data. Notably, on the RND dataset, training with just 1% of the data achieves remarkably high returns, exceeding the full-data baseline.

We further investigate the robustness of our method on datasets with highly limited state coverage using the *point mass maze* environment from ExoRL. Figure 9 visualizes the trajectories from reduced datasets collected via DIAYN and Proto strategies (*1% of DIAYN, 7% of Proto*). Compared to Figure 2 in [62], our settings exhibit even narrower state support. For example, the DIAYN dataset shows sparse trajectories near the *top-right goal*, while the Proto dataset shows limited coverage near the *bottom-right goal*. To assess performance under such constrained coverage, we evaluate

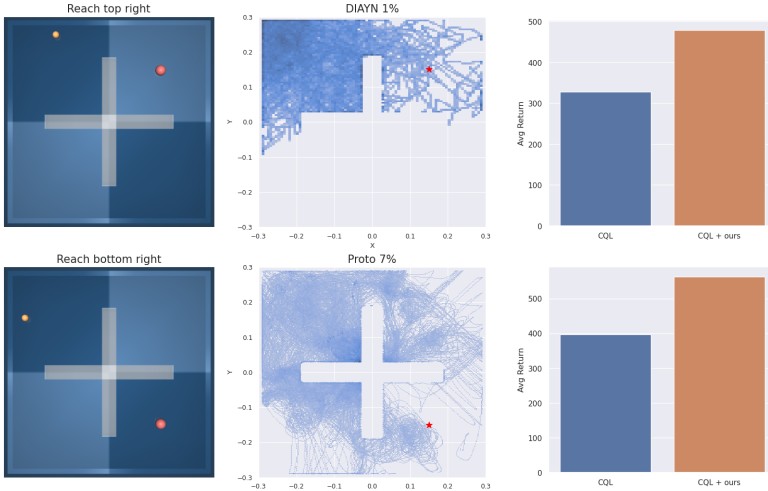

Figure 9: **Effectiveness of our method on datasets with narrow state coverage.** (Left) Visualization of goal-reaching agents and their trajectories under different goals, dataset fractions, and exploration strategies. (Right) Average episode returns of CQL trained with two datasets, with and without our pretraining method. Our approach yields significant performance gains even when the available data has limited state coverage.

CQL with and without our pretraining method on these datasets for both short-horizon (*reach top-right*) and long-horizon (*reach bottom-right*) goals. As shown in Figure 9, our method significantly improves performance even under severe state-distribution limitations. These results collectively demonstrate that our approach achieves strong data efficiency and remains effective across diverse and distributionally shifted offline datasets.

## 5  Conclusion

In this paper, we propose a simple yet effective data-efficient offline reinforcement learning method that pretrains a shared $Q$-network through a transition prediction task. The proposed framework leverages a shared network architecture that jointly predicts the next state and the $Q$-value, allowing efficient feature reuse between the transition model and value function. This design makes our approach easily applicable to a wide range of existing offline RL algorithms and substantially improves data efficiency, maintaining strong performance even when trained on limited data.

To verify the effectiveness of our method, we analyzed it under the framework of the projected Bellman equation and performed extensive experiments across diverse offline RL benchmarks, including D4RL, Robomimic, and V-D4RL. The results demonstrate that our approach consistently enhances the performance of existing offline RL methods. Furthermore, evaluations on reduced datasets and shifted data distributions confirm that our method is robustly data-efficient across varying data qualities and distributions.

**Limitations & Future Works.** This study focuses on standard model-free offline RL settings, where popular algorithms primarily rely on $Q$-function learning. Consequently, our design is tailored to complement such architectures. Nonetheless, prior works [51, 58] suggest that offline pretraining can be beneficial in broader contexts, such as unsupervised learning or goal-conditioned RL. Owing to its simplicity and plug-and-play compatibility, our method has strong potential for broader applications. Future research will extend our framework to more general settings, including offline-to-online RL, goal-conditioned RL, and real-world control scenarios.

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

# A Comparison over Sample Efficiency and Data Efficiency

In this section, we further expand the discussion on comparing the sample efficiency and data efficiency. In both online and offline RL, learning an optimal policy from limited experience data remains a fundamental challenge. Agents in both paradigms aim to extract maximal information from a finite set of interactions with the environment. This shared motivation arises from the practical constraints of data acquisition, as gathering experience in real-world or high-fidelity simulation environments is expensive and time-consuming. Consequently, both settings emphasize effective data utilization through methods such as representation learning [50, 51, 66, 60], model-based RL [22, 21, 28, 68], and off-policy optimization [20, 18], which aim to accelerate learning without proportional increases in data volume. In essence, both *sample efficiency* in online RL and *data efficiency* in offline RL quantify how effectively an agent transforms its available experience—whether gathered online or provided offline—into improved decision-making performance.

Despite this shared goal, their underlying desiderata differ substantially. In online RL, an agent alternates between two intertwined phases: experience gathering and policy update. The policy controls the distribution of collected data, which in turn affects subsequent policy updates. As a result, the experience distribution is non-stationary and tightly coupled with the evolving policy, making it theoretically and practically challenging to analyze or control [49, 45]. In contrast, offline RL learns from a fixed dataset collected by a separate behavior policy, where data distribution is static but often limited in coverage. While online RL suffers from issues such as incremental network updates and weak inductive biases [5], offline RL must contend with distribution shift and extrapolation errors [37], which hinder generalization beyond the support of the dataset.

# B Extended Discussion on Methodological Connections

In this section, we address potential concerns regarding the novelty of our method, given its conceptual connection to several prior approaches in related research areas. We provide detailed comparisons and clarifications across two primary themes: representation learning and model-based reinforcement learning.

**Representation Learning.** Recent years have witnessed a surge of research on predictive representations in reinforcement learning. Our approach—pretraining a shared Q-network through a next-state prediction task—shares the spirit of prior work on representation learning for improving data efficiency [50, 19], yet differs in key methodological aspects.

Specifically, Schwarzer et al. [50] proposed an online self-supervised pretraining scheme based on latent-space prediction, relying heavily on auxiliary design choices such as data augmentation [65] and target encoders [26]. In contrast, our method adopts a supervised pretraining objective directly on the next-state prediction task, avoiding such additional mechanisms. This simplicity enables our approach to be seamlessly integrated with diverse offline RL algorithms while consistently improving data efficiency and performance across locomotion and manipulation benchmarks.

Similarly, Guo et al. [19] introduced an unsupervised belief-state encoder for partially observable settings (POMDPs). Their focus lies in inferring the hidden state from a trajectory using a recurrent GRU-based network that predicts future observations. In contrast, our approach operates in the fully observable MDP setting using a simple MLP-based architecture that predicts the next state $s_{t+1}$ from the current state–action pair $(s_t, a_t)$. Thus, while both approaches leverage predictive modeling, our contribution lies in unifying this principle with offline RL through a shared $Q$-network architecture that enhances data efficiency without sequential modeling or partial observability assumptions.

**Model-based RL.** While our method employs a transition prediction objective, its design philosophy and application differ fundamentally from conventional model-based RL. Classical model-based approaches [54] explicitly use the learned dynamics for planning or policy improvement, whereas our approach leverages transition prediction solely for representation pretraining.

Recent methods such as TD-MPC [25] and TD-MPC2 [24] integrate model-based objectives by recursively feeding outputs of a shared encoder for transition and value learning. Our approach instead introduces a dueling-style shared architecture [57], where distinct shallow heads are used for the transition model and $Q$-value estimation. Moreover, we propose a two-phase training scheme: first, a transition-prediction pretraining phase that shapes the shared backbone; and second, a standard

RL training phase that fine-tunes the $Q$-network initialized from the pretrained backbone. This staged training framework reduces complexity and training cost while yielding substantial data efficiency gains.

JOWA [10] also employs shared Transformer-based backbones for multi-task offline RL through sequence modeling. However, while JOWA focuses on scaling across tasks and environments with few-shot fine-tuning, our objective is to improve data efficiency in conventional single-task offline RL without additional architectural or training overhead.

Dreamer [23] further advanced model-based RL with sophisticated latent world models and reconstruction objectives for planning. Although highly effective, such designs introduce considerable computational and data requirements. In contrast, our method provides a lightweight, plug-and-play alternative that enhances sample efficiency within existing offline RL frameworks, offering a practical balance between architectural simplicity and empirical performance.

**Summary.** In essence, our work departs from both model-based and representation-learning paradigms by introducing a shared $Q$-network architecture with two-phase supervised pretraining. This simple yet powerful mechanism enhances feature reuse, reduces approximation error, and consistently improves data efficiency—without the need for auxiliary objectives, sequential modeling, or complex multi-stage optimization.

## C Implementation Details

This section provides the detailed implementation setup used in our experiments. Since our proposed method is a plug-and-play pretraining approach applicable to popular offline RL algorithms, we build directly on open-source implementations for fair and consistent comparisons. Specifically, we adopt publicly available PyTorch-based repositories for each baseline:

- D4RL
    - AWAC[4]
    - CQL[5]
    - IQL[6]
    - TD3+BC[7]
- Robomimic
    - Official Robomimic repository for all baselines [8]

We restrict our comparisons to PyTorch-based baselines to ensure implementation consistency.

For D4RL experiments, each agent is trained for 1 million gradient steps per environment across five random seeds. Evaluation is conducted every 5k gradient steps for AWAC, CQL, and TD3+BC, and every 10k steps for IQL, using five rollouts per evaluation. For Robomimic experiments, each agent is trained for 200k gradient steps per environment and evaluated using 50 rollouts across five seeds. All reported results in tables and figures correspond to the best evaluation scores achieved during training. All experiments were conducted on a single NVIDIA RTX A5000 GPU for both training and evaluation.

## D Tasks and Datasets

In this section, we describe the experimental setups for the tasks and datasets used in our study. The corresponding environments are illustrated in Figure 10.

---

[4]`https://github.com/hari-sikchi/AWAC`
[5]`https://github.com/young-geng/CQL`
[6]`https://github.com/Manchery/iql-pytorch`
[7]`https://github.com/sfujim/TD3_BC`
[8]`https://github.com/ARISE-Initiative/robomimic`

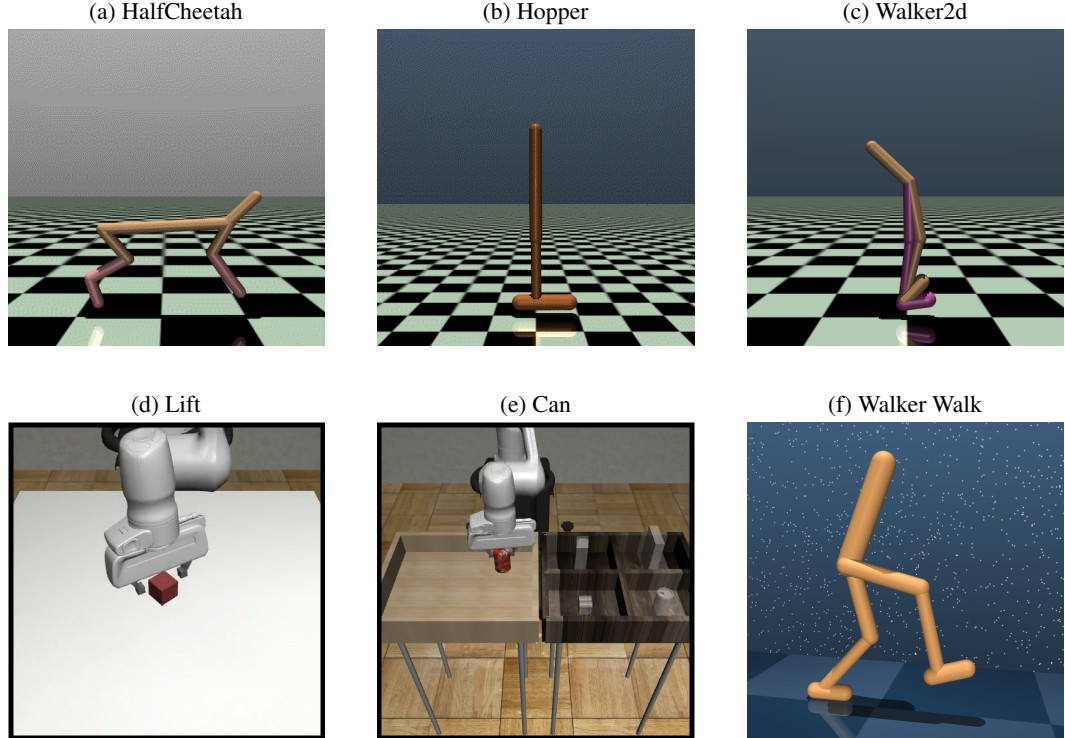

Figure 10: **Illustrations of tasks.** (a-c) Locomotion tasks on the OpenAI Gym MuJoCo and D4RL benchmark; (d-e) Tabletop manipulation tasks on the Robomimic benchmark; (f) Locomotion task on the ExoRL benchmark.

## D.1    D4RL

The D4RL benchmark consists of eight distinct task families. For our main experiments, we focus on the OpenAI Gym MuJoCo continuous control suite, which includes four environments: HalfCheetah, Walker2d, Hopper, and Ant.

Each environment provides five datasets that differ in data quality and collection strategy:

- Random (1M samples): Collected using a randomly initialized policy.
- Expert (1M samples): Collected from a policy fully trained with SAC.
- Medium (1M samples): Collected from a partially trained policy, achieving roughly one-third of the expert's performance.
- Medium-Expert ( 2M samples): A 50-50 combination of the medium and expert datasets.
- Medium-Replay ( 3M samples): Collected from the replay buffer of the medium-level agent during training.

All environments have an episode horizon of 1000 steps, and each agent's objective is to maximize forward velocity while avoiding instability. More details can be found in the official D4RL repository: https://github.com/Farama-Foundation/D4RL.

## D.2    Robomimic

The Robomimic benchmark provides a large and diverse collection of robotic manipulation demonstrations, including both human and machine-generated data of varying quality. In our experiments, we use the machine-generated (MG) datasets, which are produced by training SAC agents on each task and saving demonstrations from intermediate checkpoints to obtain mixed-quality data. We select

these datasets because our method consistently demonstrates strong performance on suboptimal data, as observed in D4RL. Each environment has an episode length of 400 steps. The Lift task requires the robot to lift a cube above a designated height, whereas the Can task requires placing a can into the appropriate container. More details are available at: https://github.com/ARISE-Initiative/robomimic.

### D.3 ExoRL

The ExoRL benchmark provides exploratory datasets for six domains from the DeepMind Control Suite: Cartpole, Cheetah, Jaco Arm, Point Mass Maze, Quadruped, and Walker, comprising a total of 19 tasks. Each dataset is collected using nine unsupervised RL algorithms—APS, APT, DIAYN, Disagreement, ICM, ProtoRL, Random, RND, and SMM—implemented in the Unsupervised Reinforcement Learning Benchmark (URLB), each trained for 10 million steps. Further information can be found in the official ExoRL repository: https://github.com/denisyarats/exorl?tab=readme-ov-file.

## E   Experiments in Dexterous Manipulation

We further evaluate our method on the Adroit benchmark from D4RL [15], to examine its applicability to more complex domains—specifically, dexterous manipulation. An illustration of the Adroit environment is shown in Figure 11. The Adroit domain involves controlling a 24-DoF robotic hand to perform four distinct manipulation tasks: Pen, Door, Hammer, and Relocate. Each task provides three datasets of varying quality:

- Human: 25 expert demonstrations collected from human teleoperation, as provided in the DAPG repository [48].
- Cloned: A 50-50 combination of human demonstrations and 2,500 trajectories generated by a behavior-cloned policy trained on the demonstrations. The demonstration trajectories are duplicated to match the number of cloned trajectories.
- Expert: 5,000 trajectories collected from an expert policy that successfully solves each task, also provided in the DAPG repository.

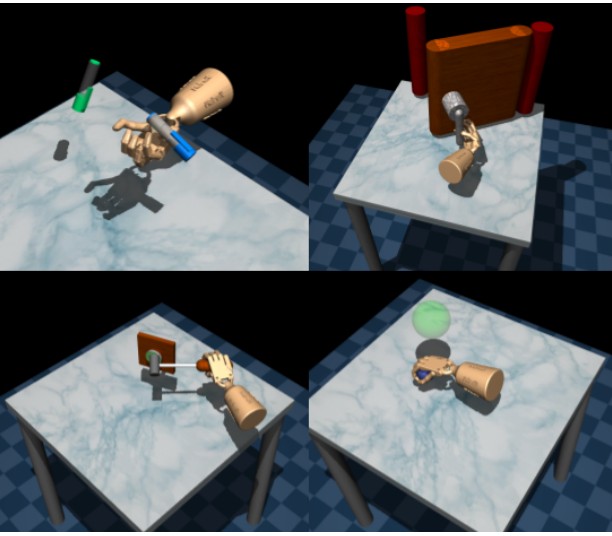

Figure 11: **Dexterous manipulation tasks of Adroit hands. (Top-left)** Pen - aligning a pen with a target orientation; **(Top-right)** Door - opening a door with a door handle; **(Bottom-left)** Hammer - hammering a nail into a board; **(Bottom-right)** Relocate - moving a ball to a target position.

For evaluation, we compare AWAC, IQL, and TD3+BC, with and without our pretraining method, across five random seeds. Table 4 reports the averaged normalized scores for each task. Across all three algorithms, integrating our pretraining phase consistently improves performance. These

results demonstrate that our approach generalizes effectively to complex, high-dimensional domains—extending beyond tabletop manipulation to dexterous hand control tasks.

Table 4: **Average normalized scores on Adroit.** Each column corresponds to a different RL baseline. The values on the left represent the baseline scores reported in the original literature, while the values on the right show the results of our method combined with each baseline. Performance improvements over the original baselines are highlighted in blue. All results are reported with the mean and standard deviation scores over five random seeds.

|  |  | AWAC | IQL | TD3+BC |
|---|---|---|---|---|
| Human | Pen | 146.19±5.29→157.60±5.28 | 101.87±14.34→104.66±17.30 | 20.32±5.97→20.78±10.93 |
|  | Hammer | 7.98±9.41→36.95±35.13 | 14.33±5.22→17.78±9.27 | 2.40±0.16→2.38±0.17 |
|  | Door | 60.82±12.38→29.96±22.43 | 6.74±1.31→5.81±3.20 | -0.09±0.00→-0.04±0.04 |
|  | Relocate | 1.51±1.05→3.91±2.21 | 1.20±1.05→1.52±1.11 | -0.29±0.01→-0.18±0.13 |
| Cloned | Pen | 145.37±4.19→144.48±3.42 | 98.38±16.13→97.76±16.90 | 39.69±18.95→48.18±11.27 |
|  | Hammer | 10.37±7.88→12.61±8.66 | 8.94±2.07→11.38±4.46 | 0.59±0.17→1.17±0.61 |
|  | Door | 2.95±2.97→9.59±7.73 | 5.61±3.02→5.00±1.44 | -0.23±0.11→-0.03±0.03 |
|  | Relocate | 0.04±0.09→0.18±0.21 | 0.91±0.45→1.06±0.40 | -0.02±0.09→-0.13±0.09 |
| Expert | Pen | 163.99±1.19→163.73±1.88 | 148.38±2.46→147.79±3.06 | 131.73±19.15→141.10±10.28 |
|  | Hammer | 130.08±1.30→130.04±0.48 | 129.46±0.42→129.50±0.36 | 33.36±34.61→59.76±52.35 |
|  | Door | 106.67±0.28→106.95±0.16 | 106.45±0.29→106.71±0.28 | 0.99±0.83→0.87±1.48 |
|  | Relocate | 109.70±1.32→111.27±0.35 | 110.13±1.52→109.82±1.45 | 0.57±0.33→0.22±0.13 |
| Total |  | 885.67±47.35→907.26±87.94 | 732.40±48.27→738.79±59.23 | 229.03±80.40→274.08±87.49 |

# F   RL Training with Pretrained Frozen Backbone Feature Extractor

In this section, we investigate the effect of shallow linear heads of the shared network architecture. Specifically, we pretrained TD3+BC and subsequently froze all network parameters except for the final linear layer during the RL training stage. The blue-colored entries in Table 5 indicate performance improvements relative to the original TD3+BC results.

Interestingly, even when only the final linear layer was trained and the shared backbone remained frozen, the model achieved higher performance than the vanilla CQL baseline. Furthermore, this frozen variant consistently outperformed others on suboptimal datasets (i.e., random, medium, and medium-replay), suggesting that the pretrained shared representation captures sufficiently rich features for effective downstream value learning, even without full fine-tuning.

Table 5: **Average normalized scores of RL training with the frozen backbone on the D4RL benchmark.** Performance improvements over the original baselines are highlighted in blue. All results are reported with the mean and standard deviation scores over five random seeds.

|  |  | AWAC | CQL | IQL | TD3+BC | freezed TD3+BC |
|---|---|---|---|---|---|---|
| Random | HalfCheetah | 2.6 | 21.7 | 10.3 | 10.2±1.3 | 6.03±2.65 |
|  | Hopper | 28.6 | 10.7 | 9.4 | 11.0±0.1 | 11.59±10.56 |
|  | Walker2d | 7.8 | 2.7 | 7.9 | 1.4±1.6 | 7.18±0.58 |
| Medium | HalfCheetah | 48.4 | 37.2 | 46.6 | 42.8±0.3 | 42.64±1.19 |
|  | Hopper | 88.4 | 44.2 | 76.9 | 99.5±1.0 | 67.16±3.56 |
|  | Walker2d | 53.0 | 57.5 | 83.8 | 79.7±1.8 | 72.03±0.78 |
| Medium Replay | HalfCheetah | 46.1 | 41.9 | 43.4 | 43.3±0.5 | 40.21±0.79 |
|  | Hopper | 101.3 | 28.6 | 96.2 | 31.4±3.0 | 64.41±19.54 |
|  | Walker2d | 88.1 | 15.8 | 77.9 | 25.2±5.1 | 41.02±12.05 |
| Medium Expert | HalfCheetah | 76.4 | 27.1 | 94.8 | 97.9±4.4 | 47.35±8.73 |
|  | Hopper | 113.0 | 111.4 | 101.8 | 112.2±0.2 | 95.07±15.27 |
|  | Walker2d | 103.3 | 68.1 | 111.6 | 101.1±9.3 | 74.75±0.59 |
| Expert | HalfCheetah | 94.4 | 82.4 | 96.4 | 105.7±1.9 | 61.93±10.71 |
|  | Hopper | 112.8 | 111.2 | 113.1 | 112.2±0.2 | 113.13±0.39 |
|  | Walker2d | 110.4 | 103.8 | 110.7 | 105.7±2.7 | 57.14±44.96 |

## G Learning Curves

In this section, we provide the full learning curves in Section 4.1 for further insights.

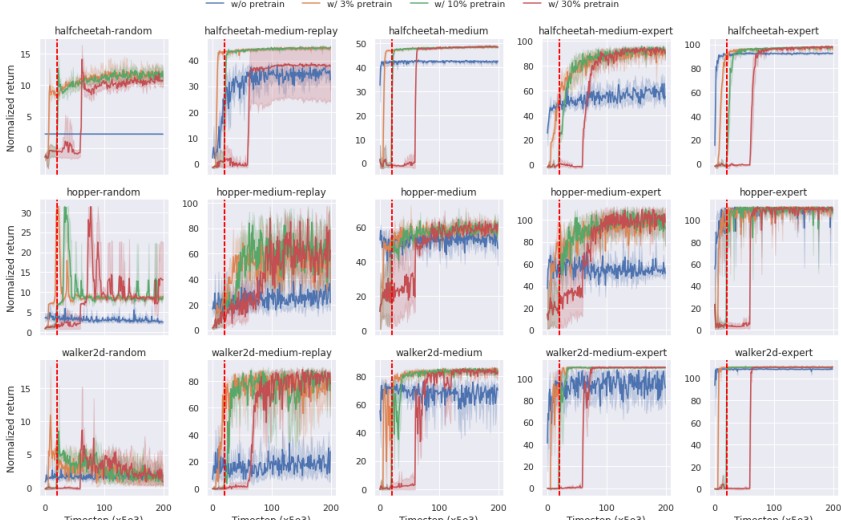

Figure 12: **Learning curves of TD3+BC on the D4RL benchmark.** We represent the normalized scores of the vanilla TD3+BC and TD3+BC with our method on progressively reduced datasets *(3%, 10%, 30%)*, respectively. The vertical red dashed lines indicate the transition between the pretraining and main training phases.

## H Rank of Latent Space during the Learning Time

We further depict the rank of the latent feature space across tasks and datasets in Section 3.1 for a comprehensive view.

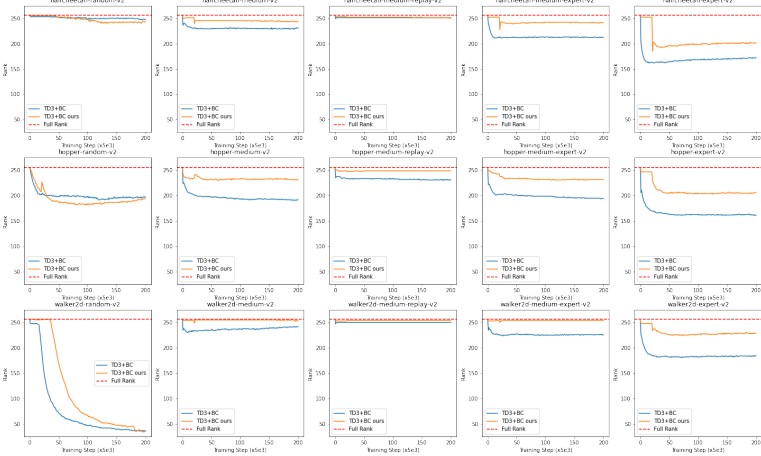

Figure 13: **Rank of the latent feature space of $Q$-network during the entire training.** We provide the rank of the vanilla TD3+BC and TD3+BC with our method, respectively. The horizontal red dashed lines stand for the full rank of the latent feature space.

# I Experiments with Varying Data Qualities and Sizes

This section provides more details on ablating the data quality and size, which is an extension of Section 4.2. All experimental results are reported with the mean and standard deviation normalized scores over five random seeds, following the same configuration in Section 4.1.

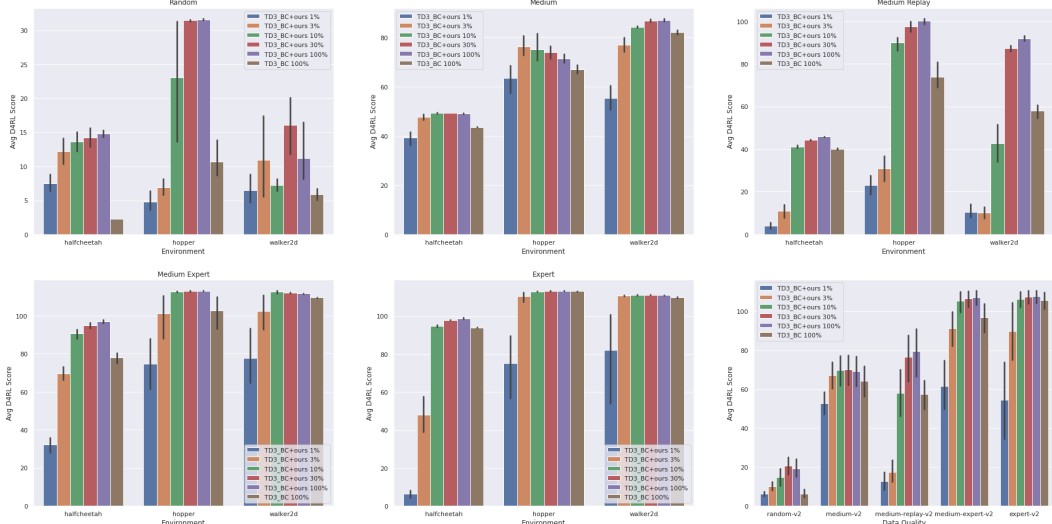

Figure 14: **Average normalized scores across dataset optimal quality and sizes.** We compare the performance of our method with TD3+BC in progressively reduced datasets *(i.e., 1%, 3%, 10%, 30%, 100% of each dataset)* to vanilla TD3+BC across the data qualities *(i.e., random, medium, medium replay, medium expert, expert)* on D4RL. Aggregated results (Bottom Right) suggest that our method guarantees better performance even in 10% of the datasets regardless of the data quality of the dataset.

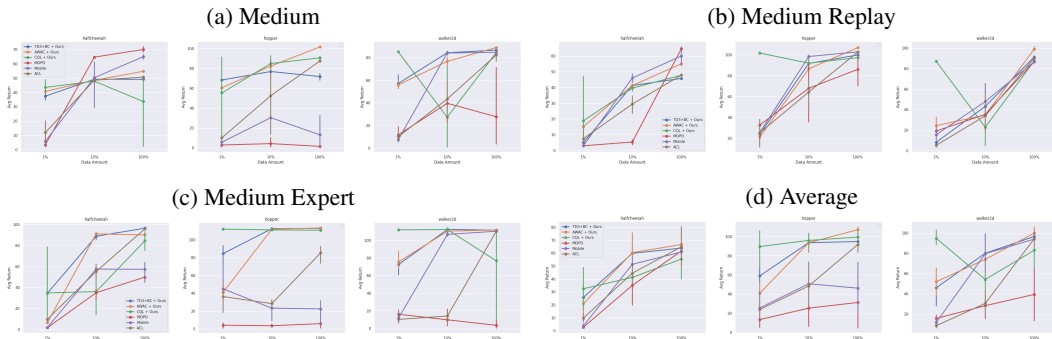

Figure 15: **Comparison with offline model-based RL and representation approaches.** We compare TD3+BC, AWAC, CQL with ours to offline model-based RLs *(i.e., MOPO, Mobile)* and a representation RL *(i.e., ACL)* on D4RL over three seeds. The graph shows results over *medium, medium-replay, medium-expert* datasets. The results show that our method maintains the performance in reduced datasets, especially 1%, unlike the other approaches.

Table 6: **Average normalized scores of AWAC across dataset sizes and qualities.**

|  |  | w/o pretrain | w/ pretrain, 10% | w/ pretrain, 30% | w/ pretrain, full |
|---|---|---|---|---|---|
| Random | HalfCheetah | 2.6 | 9.71±3.08 | 36.37±1.47 | 51.10±0.89 |
|  | Hopper | 28.6 | 97.05±3.24 | 93.35±6.32 | 59.47±33.79 |
|  | Walker2d | 7.8 | 8.57±0.47 | 8.36±1.30 | 13.11±3.91 |
| Medium | HalfCheetah | 48.4 | 55.47±1.52 | 56.64±2.68 | 54.63±1.45 |
|  | Hopper | 88.4 | 101.28±0.78 | 101.32±0.20 | 101.73±0.20 |
|  | Walker2d | 53.0 | 95.14±1.46 | 91.38±1.37 | 89.51±0.88 |
| Medium Replay | HalfCheetah | 46.1 | 51.00±0.69 | 52.12±0.76 | 55.75±1.30 |
|  | Hopper | 101.3 | 103.67±1.81 | 107.69±1.71 | 106.67±0.59 |
|  | Walker2d | 88.1 | 104.10±1.57 | 105.42±1.97 | 100.31±2.11 |
| Medium Expert | HalfCheetah | 76.4 | 83.18±1.69 | 86.55±0.94 | 90.05±1.89 |
|  | Hopper | 113.0 | 113.01±0.71 | 113.34±0.09 | 113.23±0.22 |
|  | Walker2d | 103.3 | 117.26±1.77 | 114.68±2.18 | 111.88±0.28 |
| Expert | HalfCheetah | 94.4 | 91.54±1.04 | 93.46±0.54 | 93.48±0.11 |
|  | Hopper | 112.8 | 113.02±0.17 | 113.18±0.20 | 112.86±0.10 |
|  | Walker2d | 110.4 | 117.92±2.07 | 112.55±0.56 | 111.22±0.35 |

Table 7: **Average normalized scores of IQL across dataset sizes and qualities.**

|  |  | w/o pretrain | w/ pretrain, 10% | w/ pretrain, 30% | w/ pretrain, full |
|---|---|---|---|---|---|
| Random | HalfCheetah | 10.3 | 6.92±0.63 | 12.65±2.53 | 18.28±1.02 |
|  | Hopper | 9.4 | 8.17±0.54 | 9.93±1.19 | 10.67±0.41 |
|  | Walker2d | 7.9 | 8.26±0.64 | 9.08±0.96 | 8.88±0.71 |
| Medium | HalfCheetah | 46.6 | 46.51±0.18 | 47.87±0.21 | 48.85±0.16 |
|  | Hopper | 76.9 | 75.72±3.23 | 80.76±3.51 | 78.62±2.21 |
|  | Walker2d | 83.8 | 82.62±1.03 | 83.89±1.69 | 83.63±1.14 |
| Medium Replay | HalfCheetah | 43.4 | 33.49±1.26 | 41.16±0.50 | 45.48±0.17 |
|  | Hopper | 96.2 | 80.59±8.25 | 91.08±3.67 | 99.43±1.71 |
|  | Walker2d | 77.9 | 39.08±10.42 | 75.33±4.17 | 87.95±1.68 |
| Medium Expert | HalfCheetah | 94.8 | 87.44±2.52 | 93.66±0.46 | 95.25±0.14 |
|  | Hopper | 101.8 | 93.89±10.67 | 91.05±18.78 | 105.77±11.31 |
|  | Walker2d | 111.6 | 111.23±0.83 | 111.65±0.93 | 112.09±0.93 |
| Expert | HalfCheetah | 96.4 | 77.85±3.82 | 95.88±0.44 | 97.40±0.13 |
|  | Hopper | 113.1 | 109.16±3.25 | 112.85±1.30 | 113.34±0.46 |
|  | Walker2d | 110.7 | 113.76±2.55 | 112.53±1.35 | 112.80±1.08 |

In addition to depicting results on varying dataset qualities and sizes, we numerically compare the performance of baselines for a comprehensive view in Table 6 and Table 7.

