# OpenReview forum: "Pretraining a Shared Q-Network for Data-Efficient Offline Reinforcement Learning"
_NeurIPS.cc/2025/Conference — NeurIPS 2025 poster_

### Official Review · Reviewer_XMkN · 2025-06-07

**Clarity:** 4
**Significance:** 3
**Originality:** 2
**Rating:** 5
**Confidence:** 3

**Summary:**

This paper proposes to pre-train the representation of a Q-network by learning a transition model before running model-free offline RL.
The authors show that this representation pre-training improves the performance of various model-free offline RL algorithms across multiple benchmarks, including Robomimic, D4RL, and V-D4RL.
The authors also provide a preliminary analysis showing that the latent space learned through pre-training has a higher rank, which may contribute to better Q-learning.

**Questions:**

**Q1. Q-error for Figure  3**
* Would it be possible to include the plot of Q-error to support the claim?
* Alternatively, if the rank itself is the more critical factor, it would * be helpful to clarify why it serves as a meaningful proxy for Q-error in this context.

**Q2. Clarification on Figure 8**
* Could you include the results of model-free baselines (e.g., TD3+BC, AWAC) in Figure 8?
* This would help clarify whether the observed gains in sample efficiency stem from the proposed pre-training method or simply from the underlying model-free algorithms.

**Q3. Details on the experiment of Section 3.1**
* Could you elaborate on how the rank of the latent space is computed?
* Are you following a methodology used in prior works (e.g., effective rank used in [4])?
* Providing more detail would help to understand the results and make it reproducible.

[4] Aviral Kumar et al., DR3: Value-Based Deep Reinforcement Learning Requires Explicit Regularization, ICLR 202

**Ethical Concerns:**

["NO or VERY MINOR ethics concerns only"]

**Final Justification:**

The paper introduces a modular approach, which pretraining a representation with world model training.
The concern on the discrepancy between the rank analysis and the actual Q-error was resolved.

I think this will impact offline RL (especially offline MBRL) areas on understanding how world model can benefit policy learning and Q-learning.

**Limitations:**

yes

**Quality:**

3

**Strengths And Weaknesses:**

**Strengths**

**S1. Strong Empirical Results**
* The proposed method consistently improves the performance across a range of offline RL algorithms and benchmarks.
* Notably, it also demonstrates strong gains on complex tasks, such as Robomimic and pixel-based V-D4RL.
* In addition, the method significantly improves sample efficiency; on the 1% D4RL datasets, the method outperforms prior model-based approaches by 20%–40%.

**S2. Simplicity**
* The approach is simple and modular---it can be easily plugged into existing offline RL pipelines.
* Moreover, it does not introduce significant computational overhead during training nor inference.

**S3. Writing**
* The paper is easy to read.

&nbsp;

**Weaknesses**

**W1. Novelty**
* The idea of learning representations through model-based pre-training have been actively explored (e.g., [1], [2], [3]).
* However, it is also true that its application in the context of offline RL remains underexplored.

**W2. About the analysis in Section 3.1**
* The analysis claims that to reduce the Q-error, the column space of the feature vectors and the Q-values should be closer, so having higher rank on the feature vector is important.
* Why do we need to indirectly illustrate the rank of the feature vectors, instead of directly showing the Q-error itself?
* Please correct me if I’m wrong for this point.

[1] Max Schwarzer et al., Pretraining Representations for Data-Efficient Reinforcement Learning, NeurIPS 2021
[2] Zichen Jeff Cui et al., In-Domain Dynamics Pretraining for Visuo-Motor Control, NeurIPS 2025
[3] Quentin Garrido et al., Learning and Leveraging World Models in Visual Representation Learning, Arxiv 2024.

---

> ### Author Rebuttal · Authors · 2025-07-30
>
> ## **Weakness 1:**
> >Reviewer Comment:
> Pretraining representations w/ model has been explored ([1,2,3]),  but it is also true that its application in the context of offline RL is underexplored.
>
> **Our Response:**
>
> We are grateful to the reviewer for pointing out the novelty of our method and the analysis in Section 3.1. We agree that the model-based self-predictive pretraining strategy has been widely investigated under diverse domains. However, we would like to distinguish our method from the prior approaches, conceptually and technically.
>
> In SGI, DynaMo, and I-JEPA, the encoder is pretrained with self-supervised learning for downstream policy learning. **SGI** and **I-JEPA** require an auxiliary **domain-specific data augmentation** to generate the synthetic self-predictive target for representation learning, whereas **DynaMo** only requires **raw sequential observations** for training forward and inverse dynamics in embedding space. Additionally, the **Transformer architecture** is enforced for the encoder and policy head to consume the sequential data structure for DynaMo.
>
> In contrast, **our method pretrains the shared network via a forward dynamics prediction** task, which is a supervised regression task that does **not necessitate domain-specific augmentation or network architecture**.
>
> This **plug-and-play ad-hoc architecture** exhibits several advantages:
> - It boosts the practical adoption of **many offline RL algorithms**, improving the reusability **without major modifications**.
> - It **avoids to necessitate of domain-specific techniques** for improving the underlying performance.
>
> Furthermore, as the reviewer rightly notes, our empirical evaluation spans several offline RL benchmarks (D4RL, V-D4RL, Robomimic, ExoRL). Beyond coverage, however, our primary contribution lies in demonstrating **strong performance in data-constrained settings**.
>
> We introduce a **simple yet effective architecture built on a shared Q-network pre-trained via forward dynamics prediction**. Across all experiments, our method consistently surpasses baselines—even when **trained with only 10% of the data**, demonstrating **strong data-efficiency in offline RL**. To support this, we offer **theoretical justification via the Projected Bellman Equation**, which helps explain why our approach facilitates more accurate Q-value estimation.
>
> We will revise the manuscript to clearly highlight these distinctions and contributions of our method, positioning the unique novelty of domain-agnostic applicability and strong data efficiency of our pretraining method compared to model-based pretraining strategies.
>
> ---
>
> ## **Weakness 2:**
> >Reviewer Comment:
> The analysis claims that to reduce the Q-error, the column space of the feature vectors and the Q-values should be closer, so having higher rank on the feature vector is important.
> Why do we need to indirectly illustrate the rank of the feature vectors, instead of directly showing the Q-error itself?
>
> ## **Question 1:**
> >Reviewer Comment:
> Would it be possible to include the plot of Q-error to support the claim? Alternatively, if the rank itself is the more critical factor, it would * be helpful to clarify why it serves as a meaningful proxy for Q-error in this context.
>
> **Our Response:**
>
> We appreciate the reviewer’s attention to connecting our theoretical justification with empirical observations. We recognize that the current manuscript could deliver a questionable insight on understanding the relation between the rank of the feature matrix and the Q error.
>
> In fact, it is impossible to compute the true Q error through the experiment since the Q value is approximated with the neural network, and the state and action spaces are continuous.
>
> Hence, per the reviewer’s suggestion, we now include **proxy Q-error comparisons** in the **Table 1.** below. We define a proxy Q error using **the left-hand side of Equation 5** in the manuscript ($||Q_{\varphi,\theta} - Q^\pi||_\infty$). We compare the proxy Q-error of **TD3+BC (10%)** and **TD3+BC + Ours (10%)**, where the percentile stands for the amount of dataset used for training:
> - **TD3+BC (10%)**: trained on a **10% subset** of the full dataset
> - **TD3+BC + Ours (10%)**: pretrained/trained on a **10% subset** of the full dataset
>
> To compute **the optimal Q value**, $Q^\pi$ in Equation 5, we use the Q network of **TD3+BC (100%)** for each Q error, which is **trained on the full dataset**. Since the true Q value is not available in D4RL tasks, **we consider the fully trained Q network as the best optimal Q value**.
>
> Notably, **our method achieves the lowest Q-error over all tasks**. These results indicate that our approach **learns a more accurate Q-function**, enabling truly **data-efficient** offline RL.
>
> We will revise the paper to concretely support our theoretical justification by providing the rank of the matrix and the proxy Q error in Figure 3.
>
>
> **Table 1. Proxy Q-error**
> | Task                          | TD3+BC (10%) | Ours (10%) |
> |------------------------------|------------|-------------|
> | halfcheetah-medium-v2        | 503.6851   | **287.7740**  |
> | halfcheetah-medium-replay-v2 | 1032.6671  | **119.9188**  |
> | halfcheetah-medium-expert-v2 | 522.4211   | **112.2337**  |
> | hopper-medium-v2             | 536.2078   | **472.9333**  |
> | hopper-medium-replay-v2      | 320.1790   | **319.9531**  |
> | hopper-medium-expert-v2      | 437.8861   | **254.9155**  |
> | walker2d-medium-v2           | 299.1773   | **138.0320**  |
> | walker2d-medium-replay-v2    | 379.4456   | **49.9275**   |
> | walker2d-medium-expert-v2    | 392.7360   | **148.2040**  |
> ||||
>
> ---
>
> ## **Question 2:**
> >Reviewer Comment:
> Could you include the results of model-free baselines (e.g., TD3+BC, AWAC) in Figure 8?  This would help clarify whether the observed gains in sample efficiency stem from the proposed pre-training method or simply from the underlying model-free algorithms.
>
>
> **Our Response:**
>
> We thank the reviewer for constructive feedback. We will add model-free baselines (e.g., TD3+BC, AWAC) to Figure 8 in the revised manuscript.
>
> ---
>
> ## **Question 3:**
> >Reviewer Comment:
> Could you elaborate on how the rank of the latent space is computed? Are you following a methodology used in prior works (e.g., effective rank used in [4])? Providing more detail would help to understand the results and make it reproducible.
>
>
> **Our Response:**
>
> We appreciate the reviewer’s attention to contributing reproducible research. Following the protocol in DR3 [4], we compute the rank of the feature matrix $h(s,a)$ by **feeding batches of state–action pairs to the hidden layers** of the Q network. Unlike DR3, which reports soft rank (s-rank), we calculate the **hard rank** of the feature matrix.
>
> Since we use 256 hidden units and a batch size of 256, the full rank of the resulting feature matrix is 256. We will clarify this distinction in Section 3.1 and include implementation details in the Appendix for reproducibility.
>
> ---
> ### **Reference**
> [1] Max Schwarzer et al., Pretraining Representations for Data-Efficient Reinforcement Learning, NeurIPS 2021
>
> [2] Zichen Jeff Cui et al., In-Domain Dynamics Pretraining for Visuo-Motor Control, NeurIPS 2025
>
> [3] Quentin Garrido et al., Learning and Leveraging World Models in Visual Representation Learning, Arxiv 2024.
>
> [4] Aviral Kumar et al., DR3: Value-Based Deep Reinforcement Learning Requires Explicit Regularization, ICLR 2022

---

> > ### Comment · Reviewer_XMkN · 2025-08-05
> >
> > I appreciate the authors’ clarifications and the additional experiments.
> >
> > I partially agree with the clarification on the novelty. In terms of the world model pretraining, I think Dynamo required transformer because they are doing this in a harder setting (where they don't have GT actions). Still, it is surprising that simple world model pretraining being so effective.
> >
> > The Q-error analysis clearly demonstrates that world model pretraining supports more effective Q-learning. I had initially overlooked that the "true" Q-error is not directly obtainable, but I now agree that using a proxy Q-error comparison is sufficient to support the claim.
> >
> > Accordingly, I increased my score from 4 to 5.

---

### Official Review · Reviewer_KRdK · 2025-06-28

**Clarity:** 3
**Significance:** 2
**Originality:** 2
**Rating:** 4
**Confidence:** 5

**Summary:**

The paper proposes a simple yet effective pretraining method for offline reinforcement learning (Offline RL) to address the data efficiency issue in offline RL. The method initializes the feature extraction of a Q-network by pretraining a shared Q-network, which outputs predictions of the next state and Q-value. Specifically, the shared Q-network consists of a shared deep neural network layer and two separate shallow output layers, one for predicting the next state (transition model) and the other for outputting the Q-value. During the pretraining phase, the shared network is trained using a supervised regression task to predict the transition model. After pretraining, the shared network is connected to the Q-network's shallow output layer and trained with existing offline RL value learning methods.

**Questions:**

1. Why did several reported scores of baselines significantly lower than that in widely-used implementations or original papers?
2. Would axuiliary training of dynamic during the training of Q be better or worse than the method proposed in the main paper? It seems you directly add the dynamic loss without tuning the loss weight in Eq.6 in appendix. So it's hard to say which method is better.

**Ethical Concerns:**

["NO or VERY MINOR ethics concerns only"]

**Final Justification:**

The authors addressed my concern. Although the idea of leveraging dynamic model's representation is not quite novel in online RL, applying it in offline setting and improving different offline rl algorithms could be a valuable contribution.

**Limitations:**

yes

**Quality:**

2

**Strengths And Weaknesses:**

Strengths
1. The experiments are widely performed on different alogirthms and benchmarks.
2. The method is easy to follow.

Weakness
1. The method seems a self-supervised RL (SSRL) method. Therefore, I believe the authors should at least discuss the relationships between the proposed method and existing SSRL  methods such as SAC+AE[1] and CURL[2]. Besides, in existing works such as [3] have discussed the representations effectiveness in offline RL. What's the contribution of the proposed method compared with these methods?
2. Some baseline results significantly underperform compared to widely-used implementations. For instance, CORL(https://github.com/tinkoff-ai/CORL) 's CQL achieves over 90 normalized score on halfcheetah-medium-expert, yet Table 1 reports only 27.1. BCQ could achieve success rate of 91 and 75 on Lift(MG) and Can(MG) in the original Robomimic paper, but in Figure 5 the reported results are lower than 50. Such discrepancies raise questions about the proposed method's effectiveness.
3. The writing should be significantly improved. In line 43, $g$ should refer to next state prediction and $f$ should refer to Q-value. But it seems the author write them in reverse. In line 214, 'our' should be 'Our'. In title 4.2 and 4.3, the capitalization of initial letters in words are inconsistent.

[1] Improving sample efficiency in model-free reinforcement learning from image.

[2] Representation learning with contrastive predictive coding.

[3] Understanding and Addressing the Pitfalls of Bisimulation-based Representations in Offline Reinforcement Learning.

---

> ### Author Rebuttal · Authors · 2025-07-31
>
> ## **Weakness 1:**
> >Reviewer Comment:
> The method seems a self-supervised RL (SSRL) method. Relationships between the method and SSRL (e.g., SAC+AE[1] or CURL[2])? Besides, in existing works such as [3] have discussed the representations effectiveness in offline RL. What's the contribution of the proposed method compared with these methods?
>
> **Our Response:**
>
> We thank the reviewer for highlighting the connection between self-supervised learning methods and our approach. We agree that the current Related Work section could be expanded to provide a broader contextualization, which would enhance the clarity and reliability of our empirical findings.
>
> That said, we would like to clarify a possible misunderstanding regarding the nature of our method. Specifically, our approach does not fall under self-supervised learning. In our offline pretraining phase, we use a supervised forward dynamics prediction task, where the model is trained to predict the next state given a state-action pair from the replay buffer. As we discuess in Appendix F, this is a regression task at the state level, in contrast to typical self-supervised approaches such as CURL, which focus on learning representations in the latent space via contrastive objectives.
>
> Nonetheless, we fully acknowledge the importance of learning good representations. As discussed in Section 3, when the pretrained encoder is optimized to increase the rank of the feature matrix—an implicit form of learning structured representations—the resulting Q-function initialized from this encoder demonstrates significantly improved accuracy, leading to stronger downstream performance. We believe that integrating a more advanced representation learning strategy could further enhance our method’s effectiveness across diverse environments.
>
> We will revise the related work section to reflect clear distinctions from self-supervised learning and discuss more details on how our method differentiates with the conventional self-supervised learning.
>
> ---
> ## **Weakness 2:**
> >Reviewer Comment:
> Some baseline results significantly underperform compared to widely-used implementations. For instance, CORL(https://github.com/tinkoff-ai/CORL) 's CQL achieves over 90 normalized score on halfcheetah-medium-expert, yet Table 1 reports only 27.1. BCQ could achieve success rate of 91 and 75 on Lift(MG) and Can(MG) in the original Robomimic paper, but in Figure 5 the reported results are lower than 50. Such discrepancies raise questions about the proposed method's effectiveness.
>
> ## **Question 1:**
> >Reviewer Comment:
> Why did several reported scores of baselines lower than other implementations?
>
> **Our Response:**
>
> We appreciate the reviewer’s attention to the reliability and consistency of our experimental results. We acknowledge that some of the reported scores may differ from those available in public repositories.
>
> Ensuring fairness and reproducibility has been a top priority in conducting our experiments across various baselines. To this end, we built our method on top of widely used offline RL algorithms that offer official, author-maintained implementations. Below, we provide a detailed analysis of why some reproduced scores differ from those reported in the original implementations:
>
> > CORL’s CQL Performance
>
> To validate the reason for lower score, we first evaluated our method based on the CORL implementation of CQL, running experiments with three random seeds per task and comparing the resulting D4RL scores to those reported in the original CORL paper. As shown in the table below, our method consistently outperforms the baseline, validating the effectiveness of our method across different implementations.
>
> **CORL implementation: original CQL vs CQL + Ours**
>
> | **Task** | **Implementation** | **Medium** | **Medium Replay** | **Medium Expert** |
> | --- | --- | --- | --- | --- |
> | halfcheetah | original CQL | 47.04 ± 0.22 | 45.04 ± 0.27 | 95.63 ± 0.42 |
> | | CQL + Ours | **47.28 ± 0.30** | **46.10 ± 0.07** | **96.06 ± 0.08** |
> | hopper | original CQL | 59.08 ± 3.77 | 95.11 ± 5.27 | 99.26 ± 10.91 |
> | | CQL + Ours | **72.10 ± 11.55** | **101.49 ± 0.11** | **109.00 ± 1.14** |
> | walker2d | original CQL | 80.75 ± 3.28 | 73.09 ± 13.22 | 109.56 ± 0.39 |
> | | CQL + Ours | **83.83 ± 0.61** | **86.08 ± 0.76** | **110.99 ± 0.61** |
>
> Additionally, we found that our implementation differs from CORL's in terms of hyperparameters in **the table below**. For example, the CORL implementation uses a deeper network, weight initialization, and a policy learning rate of 3e-5, whereas our version follows a commonly used setting in deep RL, including a policy learning rate of 3e-4.
>
> **Comparion on the hyperparameters across implementations**
>
> | **Hyper Parameter** | **Impl. in CORL** | **Impl. in Our Paper** |
> | --- | --- | --- |
> | Policy Learning Rate | 3e-5 | 3e-4 |
> | CQL Alpha | 10.0 | 5.0 |
> | Orthogonal Init | O | X |
> | Hidden Layers | 3 | 2 |
>
> > Robomimic’s BCQ Performance
>
> According to Appendix G of the Robomimic paper (Mandlekar et al., 2021), policy performance can vary significantly depending on factors such as input modality (state vs. image), dataset quality (machine-generated vs. human-demonstrated), and task difficulty (easy vs. hard). As further shown in Table 2 of Luo et al. (2021), the originally reported scores for CQL and BCQ are often significantly higher than their reproduced counterparts.
>
> We recognize this as an important issue to address in future work. To mitigate potential confusion, we will clearly indicate which scores come from original papers and which are our reproduced results in the revised paper.
>
> **Reference**
>
> Luo, Jianlan, et al. "Action-quantized offline reinforcement learning for robotic skill learning." Conference on Robot Learning. PMLR, 2023.
>
> Mandlekar, Ajay, et al. "What matters in learning from offline human demonstrations for robot manipulation." arXiv preprint arXiv:2108.03298 (2021).
>
> ---
> ## **Weakness 3:**
> >Reviewer Comment:
> The writing should be significantly improved. In line 43,  should refer to next state prediction and  should refer to Q-value. But it seems the author write them in reverse. In line 214, 'our' should be 'Our'. In title 4.2 and 4.3, the capitalization of initial letters in words are inconsistent.
>
> **Our Response:**
>
> We appreciate the reviewer’s suggestions for improving readability. As pointed out, Line 43 lists the outputs of the shallow layers in reverse order. We thank the reviewer for catching this detail in the shared network architecture and will revise the manuscript to prevent such potential misunderstandings.
>
> > Regarding capitalization
>
> We also appreciate the reviewer’s attention to grammatical consistency. We acknowledge the presence of inconsistent expressions and capitalization in the current draft and will revise the manuscript to ensure grammatical correctness and a more professional tone throughout.
>
> ---
> ## **Question 2:**
> >Reviewer Comment:
> Would auxiliary training of dynamics during the training of Q be better or worse than the method proposed in the main paper? It seems you directly add the dynamic loss without tuning the loss weight in Eq.6 in the appendix. So it's hard to say which method is better.
>
> **Our Response:**
>
> We appreciate the reviewer’s insightful suggestion regarding the integration of the pretraining objective into downstream RL training. We agree that incorporating the forward dynamics prediction task into the learning process could offer additional potential benefits for downstream offline policy learning.
>
> To support this discussion, we present a detailed comparison between our method and a variant that incorporates the forward dynamics prediction as a regularization term during training, as described in Eq. 6:
>
> **Comparison Between Our Method and the Eq. 6 Variant**
>
> | **Data** | **HalfCheetah w/ Eq.6** | **HalfCheetah Ours** | **Hopper w/ Eq.6** | **Hopper Ours** | **Walker2d w/ Eq.6** | **Walker2d Ours** |
> | --- | --- | --- | --- | --- | --- | --- |
> | Random | 11.45 ± 0.51 | **14.8 ± 0.5** | 31.54 ± 0.42 | **31.6 ± 0.2** | **13.46 ±  6.58** | 11.2 ± 5.1 |
> | Medium | 48.23 ± 0.33 | **49.2 ± 0.3** | 70.86 ± 2.17 | **71.5 ± 2.2** | 82.65 ± 1.65 | **87.1 ± 0.6** |
> | Medium Replay | 44.93 ± 0.29 | **45.8 ± 0.3** | 90.39 ± 7.34 | **100.2 ± 1.6** | 86.11 ± 1.54 | **92.0 ± 1.6** |
> | Medium Expert | 93.55 ± 1.00 | **96.9 ± 0.9** | **113.44 ± 0.35** | 113.0 ± 0.2 | **111.88 ± 0.63** | 111.6 ± 0.4 |
> | Expert | 96.59 ± 0.25 | **98.9 ± 0.6** | 113.28 ± 0.20 | **113.4 ± 0.3** | 110.98 ± 0.22 | **111.0 ± 0.2** |
>
> These results demonstrate that our method achieves superior performance in 12 out of 15 settings, supporting our hypothesis that pretraining the shared network with a supervised regression task significantly improves downstream policy learning.
>
> In addition, our method is more computationally efficient than the Eq. 6 variant. While Eq. 6 requires computing both the TD loss and the forward dynamics loss throughout the entire training process, our method computes the forward dynamics loss only during the pretraining phase.
>
> We agree that adaptively tuning the strength of the forward dynamics loss in the combined objective (i.e., Eq. 6) may further enhance performance. However, we would like to emphasize that our key empirical insights are derived from the design choice to pretrain the shared network via the forward dynamics prediction task, other than jointly optimizing it during policy training. Hence, this line of investigation lies outside the scope of this work.
>
> We will revise the manuscript to clearly highlight the improvements of our method over the Eq. 6 baseline and better articulate this distinction.

---

> > ### Comment · Reviewer_KRdK · 2025-08-05
> >
> > Thanks for your reply. I'm sorry that I misunderstood the pretraining as a SSRL method. I appreciate the additional results to further support the effectiveness of the proposed method. Actually I like the simple yet effective method, and I hope you could include the new results into the revised paper. I also forgot to point out pretraining a dynamic model and using it's representational part have also been applied in online RL to help improve the performance[1], which could be an important supplymentary to your literature review. Thanks again for your hard work, I have raised my score to 4.
> >
> >
> > [1] Towards General-Purpose Model-Free Reinforcement Learning. Arxiv 2025.

---

### Official Review · Reviewer_9vaj · 2025-06-30

**Clarity:** 2
**Significance:** 2
**Originality:** 2
**Rating:** 5
**Confidence:** 4

**Summary:**

This paper proposes a pertaining objective for offline RL, specifically a forward dynamics surrogate loss that pretrains the critic network before TD learning. The pretraining objective mitigates loss of expressivity (shown through rank reduction in the network outputs) in the critic network cause by using TD learning from scratch. Experiments show that this pretraining objective is useful in improving the sample efficiency of existing offline RL algorithms across a range of domains. The authors highlight the their method trained with 10-50% of the dataset can match baseline offline RL algorithms trained on the whole dataset, validating their claims.

**Questions:**

From weaknesses
- Comparing with existing pretraining objectives
- Comparing on more image based tasks


General Suggestions and Questions
- They mention that they don't reconstruct the full image but instead the extracted features from a pretrained encoder. Is this encoder fine-tuned during the RL phase or is it always frozen?
- It would be helpful to aggregate performance increases (average % increase perhaps) per algorithm/task in Table 1 to provide a quick summary

**Ethical Concerns:**

["NO or VERY MINOR ethics concerns only"]

**Final Justification:**

The authors have resolved my major concerns with additional experiments and explanations. The broad applicability of the method and easy integration make it relevant to the research community.

**Limitations:**

Yes

**Quality:**

3

**Strengths And Weaknesses:**

Strengths
- The authors perform extensive experiments in state-based environments to validate their claims. They compare their surrogate objective on a range of offline RL benchmarks, showing that their approach is generally applicable.
- The experiments show strong performance uplifts across tasks.
- The experiments on comparing performance across data scales provides strong evidence supporting their claim of improved sample efficiency.
- Comparing performance under different data distributions highlights the robustness of the objective.
- The method is simple to implement, aiding in the broader adoption of their approach
- The curves showing the difference in the effective rank of the network is a good addition to the paper.


Weakness
- Lack of comparison with existing pretraining objectives: The authors do not compare with SGI [1], or ATC [2]. While they have shown that their approach is beneficial, it would be good to have a point of reference with existing literature. There is an extended discussion of related works in the appendix which I appreciate, but I do think actual experimental comparisons on some environments/tasks would be helpful. Also I disagree that SGI is an online method, since SGI has experiments with offline datasets in Atari. This is the most importance weakness that I wish the authors would address for changing my rating.
- Lack of experiments in the image domain. Observation reconstruction is much harder in image based environments compared to state based environments like D4RL. While the authors have compared two envs from V-D4RL, I would like to see at least a few more experiments in image based tasks (from V-D4RL or visual OGBench). The data scaling experiments on image based tasks would be useful too.
- The authors use the phrase "our method guarantees better performance" in Fig 7 and in the text below which is too strong I feel given that they have provided mostly empirical evidence. I would suggest they tone down that language
- Clarity in writing: The paper is a bit hard to follow at times with odd phrasing and repeated language.

[1] Schwarzer, Max, et al. "Pretraining representations for data-efficient reinforcement learning." Advances in Neural Information Processing Systems 34 (2021): 12686-12699.
[2] Stooke, Adam, et al. "Decoupling representation learning from reinforcement learning." International conference on machine learning. PMLR, 2021.

---

> ### Author Rebuttal · Authors · 2025-07-30
>
> ## **Weakness 1:**
> >Reviewer Comment:
> The authors do not compare with SGI [1], or ATC [2]. While they have shown that their approach is beneficial, it would be good to have a point of reference with existing literature.
>
> ## **Question 1:**
> >Reviewer Comment:
> Comparing with existing pretraining objectives
>
> **Our Response:**
>
> We appreciate the reviewer’s feedback for pointing out the connection between our work and existing literature with a pretraining objective in RL. We agree that the current manuscript lacks comprehensive explanations on comparing pretraining objectives in reinforcement learning, especially for offline RL. Before we outline our plans to revise the manuscript, we would like to clarify the differences between our method and the suggested pretraining objectives, which highlight the unique novelty of our approach.
>
> > Comparison with existing pretraining objective
>
> Pretraining strategy has been actively investigated in the reinforcement learning field to boost the performance of an underlying policy. The target of pretraining is often the visual feature extractor — the encoder — that extracts a latent vector from high-dimensional sensory input. SGI [1] and ATC [2] are representative methods of how the pretraining objective enables efficient downstream vision-based RL training. However, as we discuss in Appendix F, we clarify that the novelty of our method is based on its simplicity and adaptability without enforcing any modification on the base RL algorithm nor any constraint on problem formulation.
>
> Typical pretraining method for visual RL involves an unsupervised learning task (e.g., contrastive learning) that relies on a synthetic technique for generating a learning signal. For instance, ATC employs random-shift data augmentation to obtain positive and negative samples for computing InfoNCS loss, which has been proven effective in improving sample efficiency in visual RL. However, this domain-specific data augmentation technique may hinder the broader applicability of an RL algorithm by limiting the domain of the algorithm can solve. On the other hand, we demonstrate our method’s generalizability to diverse problem formulations through extensive empirical experiments, including offline, online, and visual RL. Additionally, we observe that our method improves the data efficiency of offline RL across data collection strategies and constrained distribution of state space.
>
> We will reinforce the novelty of our method in the revised paper as follows:
>
> * **We will reconstruct the related work section in the main paper** to clearly distinguish our method from self-predictive pretraining methods.
> * **We will expand the discussion about related work in Appendix F**, which offers a broader and comprehensive understanding of pretraining objectives in diverse RL formulations.
>
> We believe that these modifications will reposition the contribution of our method as an effective pretraining strategy for data efficiency.
>
> > Our argument on SGI is an online method
>
> We thank the reviewer for thorough feedback. It is indeed true that **SGI is not limited to a method that can only be applied to an online RL setting**. As the reviewer notes, SGI validates its strong performance by considering diverse sources of offline data to pretrain the encoder in the Atari-100k benchmark. To explain this misidentification, SPR is a self-predictive pretraining method for online RL, as we discuss in Appendix F, and we have confused SPR with SGI at the time of writing the manuscript. We agree that the manuscript delivers a misunderstanding of SGI and degrades the reliability of our method.
>
> We will fix this incorrectness in the introduction and related work section in the revised paper.
>
> **Reference**
> Schwarzer, Max, et al. "Data-efficient reinforcement learning with self-predictive representations." arXiv preprint arXiv:2007.05929 (2020).
>
>
> ---
> ## **Weakness 2:**
> >Reviewer Comment:
>  Lack of experiments in the image domain. Observation reconstruction is much harder in image based environments compared to state based environments like D4RL. While the authors have compared two envs from V-D4RL, I would like to see at least a few more experiments in image based tasks (from V-D4RL or visual OGBench). The data scaling experiments on image based tasks would be useful too.
> ## **Question 2:**
> >Reviewer Comment:
> Comparing on more image based tasks
>
> **Our Response:**
> We thank the reviewer for suggesting thorough comments. We agree that the current results in the image-based environment may visualize a limited context of which our method can improve the base visual RL algorithm across tasks.
>
> To address the reviewer’s concern about the image-based experiment, we conduct additional experiments on the humanoid_walk task across data quality. Additionally, we consider an extra dataset quality, medium_replay, during the additional experiment for a broader understanding of the effect of the policy optimality. The episodic return is averaged over three random seeds for each data quality in the table below.
>
> ### **Extra image-based tasks performance**
> | Environment     | Data           | DrQ-v2             | DrQ-v2 + Ours       |
> |-----------------|----------------|--------------------|---------------------|
> | walker_walk     | medium          | 306.93 ± 28.21     | **338.77 ± 29.55**  |
> | | medium_replay   | **30.09 ± 0.75**       | 28.68 ± 2.29    |
> | | medium_expert   | 352.46 ± 37.15     | **369.66 ± 20.86**  |
> | cheetah_run     | medium          | 340.33 ± 7.55      | **379.80 ± 45.83**  |
> | | medium_replay   | 21.15 ± 2.04       | **25.13 ± 2.04**    |
> | | medium_expert   | 251.52 ± 34.37     | **258.76 ± 50.33**  |
> | humanoid_walk   | medium          | 12.57 ± 6.73       | **20.03 ± 3.80**    |
> | | medium_replay   | **40.76 ± 16.27**      | 19.38 ± 6.10    |
> | | medium_expert   | 4.11 ± 2.72        | **5.12 ± 1.89**     |
> ||||
>
> In summary, our method indicates improved performance in seven out of nine cases across three tasks and three data qualities. We believe these additional results support the applicability of our method to the image-based domain.
>
> We will incorporate these empirical observations into the revised paper to make the novelty of our method more concrete.
>
> ---
> ## **Weakness 3:**
> >Reviewer Comment:
> The authors use the phrase "our method guarantees better performance" in Fig 7 and in the text below which is too strong I feel given that they have provided mostly empirical evidence. I would suggest they tone down that language.
>
>
> **Our Response:**
> We appreciate the reviewer on suggesting meticulous feedback. In Figure 7, we conduct data scale experiments to verify data efficiency of our method across various scales. We acknowledge that the current manuscript would deliver unintentionally strong tone of our argument.
> We will revise the expression “From the overall results, we conclude that our method guarantees better performance” to ”The empirical observations support the data efficiency of our method”.
>
> ---
>
> ## **Weakness 4:**
> >Reviewer Comment:
> The paper is a bit hard to follow at times with odd phrasing and repeated language.
>
> **Our Response:**
> We thank the review for their thorough comment. We apologize for some inconvenient writings. We will revise the overall manuscript to provide clear, coherent idea on our method.
>
> ---
>
> ## **Question 3:**
> >Reviewer Comment:
> They mention that they don't reconstruct the full image but instead the extracted features from a pretrained encoder. Is this encoder fine-tuned during the RL phase or is it always frozen?
>
>
> **Our Response:**
> We appreciate the reviewer on insightful question. Our method is designed to be *plug-and-play*, meaning it can be integrated into existing RL algorithms without requiring architectural modifications. To preserve this modularity, we applied our approach by altering only the Q-function loss, without changing other components of the underlying algorithm.
>
> For image-based tasks, we adopted **DrQ-v2** as the backbone RL algorithm. As described in the DrQ-v2 framework, the visual encoder is continusouly trained during the RL phase other than kept frozen. We follow this protocol and update the encoder during policy training. This design choice aligns with our goal of preserving compatibility with established pipelines while evaluating the effectiveness of our Q loss formulation.
>
> ---
> ## **Question 4:**
> >Reviewer Comment:
> It would be helpful to aggregate performance increases (average % increase perhaps) per algorithm/task in Table 1 to provide a quick summary.
>
>
> **Our Response:**
> We thank the reviewer on improving readability of the manusciprt. Through this rebuttal, we aggregate experimental results on two benchmarks, D4RL and V-D4RL. We calculate the average improvements using blow equation:
>
> > Improvement = ((before -after)/before) x 100 (%)
>
> In below tables, we average the improvements over each environment and dataset.
>
> ### **Average Improvement (%) by Task and Algorithm (D4RL)**
> |    | AWAC         | CQL          | IQL          | TD3+BC       |
> |---------------|--------------|--------------|--------------|--------------|
> | HalfCheetah   | +325.89% | +42.48%  | +15.26%  | +101.56% |
> | Hopper        | +22.13%  | +88.97%  | +4.61%  | +42.77%  |
> | Walker2d      | +28.67%  | +195.83% | +6.74%  | +36.85%  |
> | Overall Avg. | +140.37% | +138.89% | +9.72%   | +67.58%  |
> ||||||
>
> ### **Average Improvement (%) by Task (V-D4RL)**
> | Walker walk | Cheetah run | Humanoid walk |
> |-------------|-------------|----------------|
> | +3.52%   | +11.13%  | +10.49%     |
> ||||
>
> ---
> ### **Reference**
> [1] Schwarzer, Max, et al. "Pretraining representations for data-efficient reinforcement learning." Advances in Neural Information Processing Systems 34 (2021): 12686-12699.
>
> [2] Stooke, Adam, et al. "Decoupling representation learning from reinforcement learning." International conference on machine learning. PMLR, 2021.

---

> ### Author Response · Authors · 2025-08-06
>
> We appreciate the reviewer for highlighting our efforts to make a significant step during the rebuttal.
>
> As we explain in the rebuttal, we will reinforce the novelty of our method in the revised paper as follows:
>
> * **We will reconstruct the related work section in the main paper** to clearly distinguish our method from self-predictive pretraining methods.
> * **We will expand the discussion about related work in Appendix F**, which offers a broader and comprehensive understanding of pretraining objectives in diverse RL formulations.
> * **We will improve the grammar and readability of the manuscript** to provide a concise tone and comprehensive results.
> * **We will incorporate empirical observations in the image domains into the revised paper** to make the novelty of our method more concrete.
> * **We will improve the readability of Table 1** to provide concise and detailed results on empirical improvements of our method.
>
> Again, we thank the reviewer for a dedicated effort to flourish the RL community.

---

### Official Review · Reviewer_42nb · 2025-07-02

**Clarity:** 2
**Significance:** 2
**Originality:** 2
**Rating:** 4
**Confidence:** 4

**Summary:**

This paper addresses the challenge of improving data efficiency in offline reinforcement learning (RL), where agents must learn from static datasets without further environment interaction. The authors propose a simple, plug-and-play pretraining method that initializes a shared Q-network to predict both the next state and Q-value, enhancing performance across various offline RL algorithms. Empirical results on D4RL, Robomimic, V-D4RL, and ExoRL benchmarks show that the method significantly improves performance, even outperforming standard approaches using only 10% of the dataset.

**Questions:**

see weaknesses

**Ethical Concerns:**

["NO or VERY MINOR ethics concerns only"]

**Final Justification:**

Thanks to the authors for their response and clarifications; will retain my score leaning towards acceptance.

**Limitations:**

Yes

**Quality:**

3

**Strengths And Weaknesses:**

Strengths:
1	The idea of pre-training with next-state prediction is interesting and intuitively sound.
2	The proposed method significantly improves upon existing offline reinforcement learning approaches.
3	The paper provides an analysis based on the projected Bellman equation, which adds theoretical depth to the work.

Weaknesses:
1	The use of next-state prediction as a pre-training task is not novel, and the paper currently lacks important comparisons and citations. For example, prior works [1, 2] have also employed next-state prediction for pre-training or co-training policy networks. It is recommended that the authors discuss these related studies to better position their contribution. [1] Wu, Hongtao, et al. "Unleashing large-scale video generative pre-training for visual robot manipulation." arXiv preprint arXiv:2312.13139 (2023). [2] Radosavovic, Ilija, et al. "Humanoid locomotion as next token prediction." The Thirty-eighth Annual Conference on Neural Information Processing Systems (2024).
2	The paper does not include real-world experiments, which limits the evaluation of its practical applicability.
3	It is not clear what is the datasets that are used for pre-training.

---

> ### Author Rebuttal · Authors · 2025-07-30
>
> ## **Weakness 1:**
> >Reviewer Comment:
> Lack of related comparisons and citations. For example, prior works [1, 2] have also employed next-state prediction for pre-training or co-training policy networks. It is recommended that the authors discuss these related studies to better position their contribution.
>
> **Our Response:**
>
> We appreciate the reviewer’s thorough observation. It is true that our method employs a next-state prediction task for pretraining the shared network for data-efficient offline RL, which resembles how suggested works train their network. We agree that referencing such works will strengthen the related work section and better position our contribution in the broader landscape of the next state prediction task for pretraining.
>
> However, we would be grateful if we could clarify a potential misunderstanding regarding our method for pretraining the shared network with the next state prediction task. We introduce **a two-stage strategy for data-efficient offline RL** training by:
> - **Pretraining the shared network with the forward dynamics (next state) prediction** task
> - Training the weight-initialized, pretrained Q network with **the underlying RL training** task
>
> On the contrary, suggested references pretrain the generative large Transformer with the next token prediction task via an autoregressive reconstruction objective. As we discuss in Appendix F, involving auxiliary representation learning in policy learning is non-trivial since effective representation learning strategy often requires **domain and modality-specific techniques **(e.g., data augmentation, Transformer architecture). On the other hand, our method empirically demonstrates performance gains using **only 10% of the original dataset for both pretraining and RL, outperforming baselines trained on full datasets**.
>
> We will revise the manuscript to reinforce the contribution of our method by offering a broader, deeper discussion on comparing with the next token prediction task.
>
> ---
>
> ## **Weakness 2:**
> >Reviewer Comment:
> The paper does not include real-world experiments, which limits the evaluation of its practical applicability.
>
> **Our Response:**
>
> We appreciate the reviewer’s attention to expanding the applicability of our method into the real world. We select **D4RL, Robomimic, V-D4RL, and ExoRL** benchmarks–which offer **high-fidelity proxies for real-world dynamics** in the simulated environment– for the performance comparison across popular offline RL baselines:
> - D4RL: Most popular benchmark in offline RL.
> - Robomimic: Data from diverse robot executions in manipulation tasks.
> - V-D4RL: High-dimensional visual input that expands the applicability.
> - ExoRL: Non-i.i.d. collection strategies that mimic real-world variability.
>
> Across these domains, our method consistently:
> - Improves **data efficiency** (1–10% data usage)
> - **Outperforms baseline methods trained on full datasets**
>
> These findings position the potential practicality of our method under real-world constraints. Subsequently, we believe that future work could explore diverse real-world RL tasks by pretraining the Q network with our method to enable data-efficient offline RL.
>
> We will strengthen our Limitations section to explicitly discuss transfer to physical platforms and outline planned future experiments on real robots.
>
> ---
>
> ## **Weakness 3:**
> >Reviewer Comment:
>  It is not clear what is the datasets that are used for pre-training.
>
> **Our Response:**
>
> We thank the reviewer for the thoughtful comment. We agree that the current manuscript could confuse the reader about the sort of datasets used for pretraining.
> In every experiment, we **pretrain the shared Q-network on the same static dataset** that the offline RL algorithm later consumes.
> **No external data or additional transitions** are introduced at any stage.
> This design is central to our core claim of true **data efficiency**: by leveraging **simple** supervised next-state prediction, we extract more value from existing data **without expanding the dataset or raising computational requirements**.
>
> We will explicitly state in the Methodology section that pretraining and RL training share an identical dataset in the revised paper. Additionally, we will highlight this point in the Experimental Setup, emphasizing its importance for resource-constrained applications.
>
> ---
> ### **Reference**
> [1] Wu, Hongtao, et al. "Unleashing large-scale video generative pre-training for visual robot manipulation." arXiv preprint arXiv:2312.13139 (2023).
>
> [2] Radosavovic, Ilija, et al. "Humanoid locomotion as next token prediction." The Thirty-eighth Annual Conference on Neural Information Processing Systems (2024).

---

### Decision · Program_Chairs · 2025-09-17

**Decision:**

Accept (poster)

**Comment:**

The paper got a pretty positive reception overall: reviewers agreed it’s a simple but effective idea, easy to plug into existing offline RL pipelines, and the empirical gains, especially under low-data regimes, are convincing. The main criticisms centered on novelty (next-state pretraining isn’t new), missing comparisons to prior pretraining objectives (SGI, ATC, SSRL methods), and some baseline discrepancies. Reviewers also noted limited real-world or image-based experiments and suggested the writing could be cleaned up. The authors’ rebuttal addressed most of these, adding new experiments, clarifying dataset use, and contextualizing novelty, which helped shift reviewer scores upward. In short, the work is meaningful, well-executed, and broadly useful, with clear room for polishing and positioning in relation to existing literature.